# *Staphylococcus aureus* sacculus mediates activities of M23 hydrolases

Alicja Razew [1,2,3], Cedric Laguri [1], Alicia Vallet [1], Catherine Bougault [1], Magdalena Kaus-Drobek [3], Izabela Sabala [2,3] ✉ & Jean-Pierre Simorre [1] ✉

Peptidoglycan, a gigadalton polymer, functions as the scaffold for bacterial cell walls and provides cell integrity. Peptidoglycan is remodelled by a large and diverse group of peptidoglycan hydrolases, which control bacterial cell growth and division. Over the years, many studies have focused on these enzymes, but knowledge on their action within peptidoglycan mesh from a molecular basis is scarce. Here, we provide structural insights into the interaction between short peptidoglycan fragments and the entire sacculus with two evolutionarily related peptidases of the M23 family, lysostaphin and LytM. Through nuclear magnetic resonance, mass spectrometry, information-driven modelling, site-directed mutagenesis and biochemical approaches, we propose a model in which peptidoglycan cross-linking affects the activity, selectivity and specificity of these two structurally related enzymes differently.

The major structural component of the bacterial cell wall (CW) is peptidoglycan (PG). PG forms a three-dimensional matrix, termed the sacculus, that encompasses the cell cytoplasm[1–3] (Fig. 1a). In terms of composition, PG is a heteropolymer composed of conserved glycan chains of alternating β−1,4-linked *N*-acetylglucosamine (GlcNAc) and *N*-acetylmuramic acid (MurNAc) (Fig. 1a), and a stem peptide covalently anchored to MurNAc[4,5]. In gram-positive bacteria, the stem peptide is composed of four to five amino acids, which may be cross-linked with an adjacent stem. The composition and length of the cross-bridge is variable across species or even strains. The common feature of staphylococci is the presence of the pentaglycine cross-links[6] and a high degree of PG cross-linking (74–92% of stem peptides are cross-linked)[1]. PG architecture serves to alleviate internal turgor posed by the cytoplasm and therefore is essential for cell viability[7].

PG digestion is a prerequisite for bacterial cell growth and division[8] and is performed by PG hydrolases, a group of enzymes with a wide range of specificities. Glycosidases break up the glycan chains, amidases separate the MurNAc from the stem-peptide, endopeptidases operate cuts within the stem-peptide or the cross-bridge, and carboxypeptidases cleave the amide bond between the penultimate and the ultimate amino acid of the stem-peptide[9]. Over the years, this group of enzymes has been thoroughly characterised biochemically,

functionally and structurally[5,10,11]. Few experimental data and modelling study results are currently available that describe the interaction of PG hydrolases with small ligands mimicking PG fragments[12–16]. Nevertheless, not much is known about how these enzymes operate within the complex PG mesh. This is mainly because most structural biology techniques at the atomic level fail with PG due to its high heterogeneity and intrinsic flexibility.

To fill this gap, the present study focuses on the interplay between *Staphylococcus aureus* PG and two peptidases in the M23 family, LytM from *S. aureus* and lysostaphin (Lss) from *S. simulans*. Both enzymes hydrolyse the pentaglycine cross-bridge of staphylococcal PG[17–19] and share a common fold for their catalytic domain but play different biological roles in bacteria. LytM acts as an autolytic enzyme that is involved in the growth and division of *S. aureus*[20], whereas Lss is a potent antimicrobial weapon leveraged by bacteria to eliminate bacterial competitors residing in the same ecological niche[21]. Its unique properties have been the subject of multiple studies, which also aimed at using Lss as an antimicrobial[18,22,23].

M23 peptidases contain conserved HxxxD and HxH motifs, comprising residues that coordinate Zn(II) and polarise the carbonyl oxygen of the scissile amide bond, preceding the nucleophilic attack of the active site water. Active centre Zn(II) is located at the bottom of the

[1]Universite Grenoble Alpes, CNRS, CEA, Institut de Biologie Structurale, 71 avenue des Martyrs-CS10090, Grenoble cedex 9 38044, France. [2]International Institute of Molecular and Cell Biology in Warsaw, 4 Ks. Trojdena Street, 02-109, Warsaw, Poland. [3]Laboratory of Protein Engineering, Mossakowski Medical Research Institute, Polish Academy of Sciences, 5 Pawinskiego Street, 02-106, Warsaw, Poland. ✉e-mail: isabala@imdik.pan.pl; jean-pierre.simorre@ibs.fr

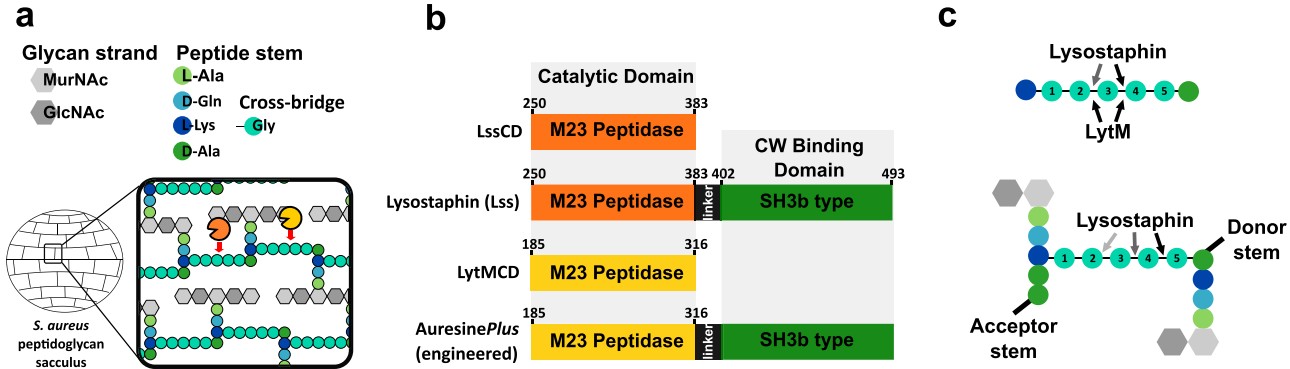

**Fig. 1 | LssCD and LytMCD hydrolyse pentaglycine within *S. aureus* PG.**
**a** Schematic representation of *S. aureus* sacculus and LssCD (orange) and LytMCD (yellow) digestion site. **b** Scheme of the PG hydrolases constructs used in the study. **c** Schematic representation of Lss and LytM hydrolytic preferences towards Gly-Gly bonds in synthetic polyglycine peptides and *S. aureus* muropeptide. For Lss, arrows are colour-coded from the most (black) to the least (pale grey) preferred hydrolysis site. Glycines were numbered as described in the text. "Donor stem-peptide" and "acceptor stem-peptide" introduce a nomenclature of the stem-peptides corresponding to the process of their cross-linking.

groove, which is limited by four loops[19]. It was postulated that the general architecture of the groove reflects differences in the specificity of M23 peptidases. For instance, Lss and LytM catalytic domains, which preferentially hydrolyse pentaglycine, display more narrow and elongated grooves compared to those of enzymes that tolerate sequence alterations in cross-bridges (e.g. LasA, EnpA)[24]. The crystal structure of LytM with the transition state analogue tetraglycine phosphinate revealed that the cleaved peptide adopts an extended, β-like structure[15]. Little is known about the interaction of LytM with more complex ligands.

The specificity of these enzymes has been defined by several different methods[25–27]. Lss is composed of catalytic domains (CD) and a cell wall binding domain (CBD), which binds the pentaglycine cross-bridge and consequently mediates enzyme specificity to *S. aureus* cells[28] (Fig. 1b). This two-domain form, which is also termed mature, is the native species of Lss. Mature Lss digests bonds of isolated pentaglycine peptides between the third-fourth and/or the second-third glycine (numbering from peptide C-terminus; Fig. 1c upper panel), whereas PG muropeptides (disaccharide stem-peptides cross-linked with a pentaglycine bridge) between the fourth-fifth, the third-fourth and the second-third glycine, with a descending order of frequency (first glycine is attached to lysine side-chain)[25,29–31] (Fig. 1c lower panel). LytMCD hydrolyses pentaglycine peptide into di- and triglycine[26], but its digestion site within muropeptide is not known. The lytic activity of both enzymes is diminished towards staphylococcal strains with cross-bridges containing glycine to serine substitution, which is a natural resistance strategy employed by bacteria against some M23 peptidases[32,33].

With this work, we intend to examine the specificity of LssCD and LytMCD for bacterial PG from a molecular basis and to produce a comprehensive view of their interactions with complex PG sacculus mesh. We use a simplified model using catalytic domains alone to gain insights into interaction of two evolutionary conserved enzymes with bacterial cell wall. To this end, we combine powerful analytical techniques, namely, ultra-performance liquid chromatography-mass spectrometry (UPLC-MS) and solution-state nuclear magnetic resonance (NMR), to obtain insights into their substrate selectivity and specificity. In addition, we use a solid-state NMR approach[34–38] to determine their interface with PG sacculus, which we illustrate with a computed, data-driven HADDOCK model[39]. Through this study, we define structural elements of these two evolutionarily related enzymes, which are important for the interaction with PG sacculus. We propose that the different interaction interfaces displayed by LssCD and LytMCD upon recognition of the PG sacculus, sensitise these enzymes to the level of PG cross-linking differently and consequently mediate their different physiological functions.

## Results

### LytMCD and LssCD hydrolyse the same bond in the muropeptide

To gain insights into the specificity and selectivity of LytMCD and LssCD, we monitored the appearance of muropeptide digestion products by solution NMR. We utilised unlabelled protein variants composed of each catalytic domain alone. Mature Lss was included in the analysis for comparison. Soluble, $^{13}$C, $^{15}$N-labelled PG fragments were generated using mutanolysin, which produces a mixture of PG muropeptide mono- and multimers (Fig. 2a, b). Muropeptide samples were then processed by Lss, LssCD and LytMCD and analysed by $^{1}$H-$^{15}$N-correlation NMR (2D $^{1}$H, $^{15}$N BEST-TROSY, Fig. 2c). NMR resonances corresponding to NH groups of the different amino acids can be observed, and cleaved glycine cross-links produce NMR signals clearly distinct from the glycines forming cross-bridge and/or anchored at the acceptor stem-peptide ($^{15}$N shifts 114–117 ppm and 107–111 ppm, respectively). This allowed us to precisely identify PG bonds cleaved by M23 peptidases.

Analysis of digestion products revealed that all studied enzymatic variants generated one predominant reaction product, namely, $G_5$-stem (Fig. 2d), which consists of the glycan strand with attached stem-peptide and the disrupted cross-bridge containing fifth glycine (hydrolysis between fourth and fifth glycine of cross-bridge; Fig. 2b, Supplementary Fig. 1). Digestion by mature and Lss catalytic domain alone produced a stem-peptide with fourth and fifth glycine attached ($G_4$-$G_5$-stem resulting from hydrolysis between third and fourth glycine of cross-bridge), in agreement with previous data reported for muropeptide digestion by Lss[31]. In addition, a minor species, corresponding to diglycine peptide ($G_2$-$G_3$/$G_3$-$G_4$), was produced by mature Lss alone. Based on previous data reporting the activity of Lss against tetraglycine, we hypothesise that this product possibly derives from further processing of a partially disrupted cross-bridge (second cut between first and second glycine or second and third glycine). LytMCD produced a triglycine peptide ($G_2$-$G_3$-$G_4$ as a secondary hydrolysis product resulting from the cleavage between first and second glycine), which has not been identified for either Lss or LssCD.

To validate these observations, we performed muropeptide digestion but followed by UPLC-MS analysis (Supplementary Fig. 2, Supplementary Table 1). To simplify the analysis, we assumed that the major digestion site is between the fourth and fifth glycine ($G_5$-stem, Fig. 2b). Then, we searched for the masses that correspond to selected potential products of the enzymes with one, two, three and four glycine residues anchored at the lysine of the donor stem. We assumed that any increase in relative abundance of peak of interest in MS spectrum above the level of the same mass readout in the undigested muropeptide sample was the product of a hydrolytic reaction with the

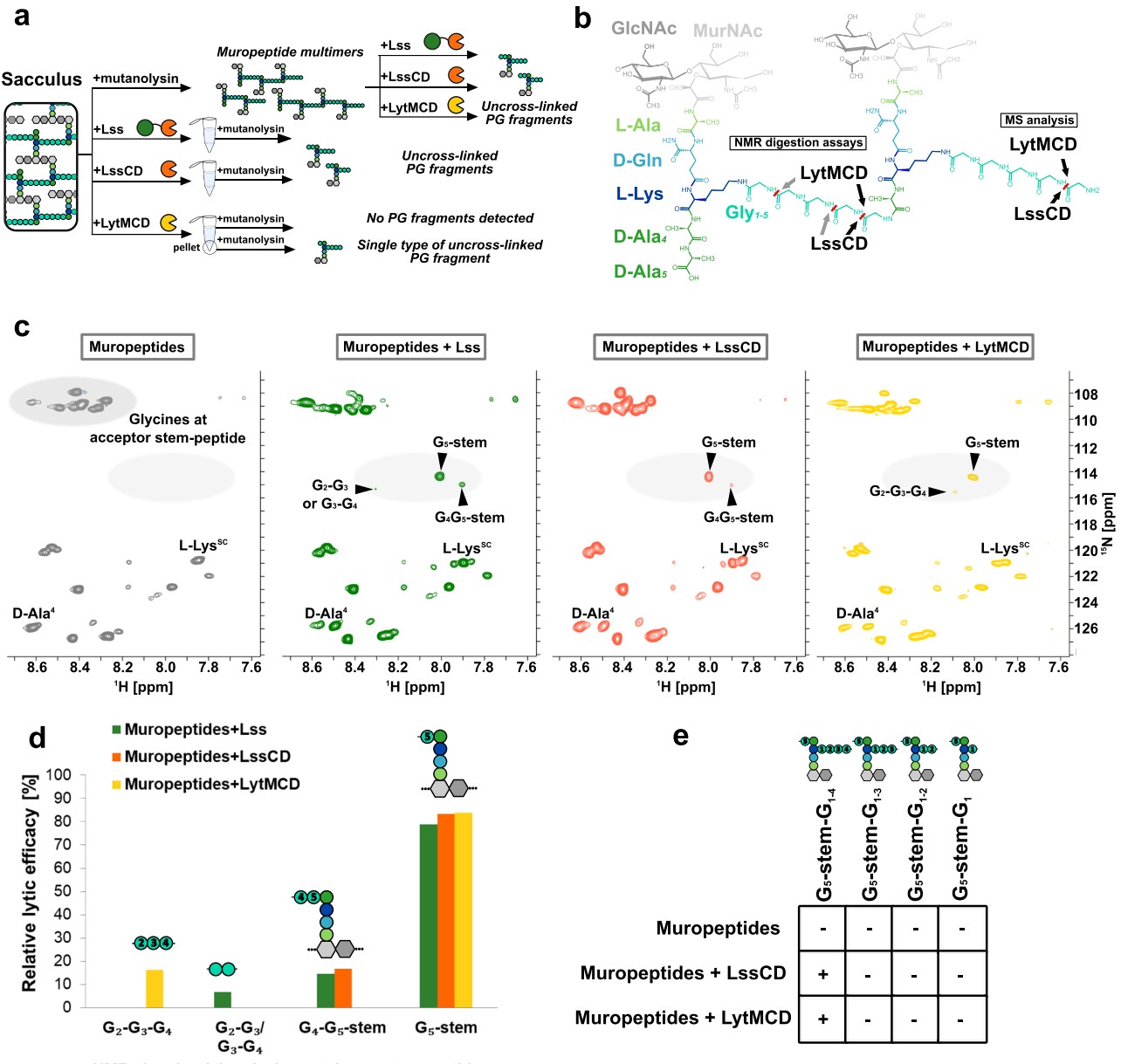

**Fig. 2 | LytMCD and LssCD hydrolyse the same bond in the muropeptide cross-bridge. a** Schematic representation of the *S. aureus* PG digestion procedures performed during the study and resulting soluble fragments. **b** Chemical structures of the muropeptide dimer. Saccharide and amino acid components are named and colour-coded as in Fig. 1a. Bonds hydrolysed by the LssCD and LytMCD defined with solution NMR and MS analysis were indicated with arrows, which are colour-coded from the most (black) to the least preferred hydrolysis site (pale grey). **c** $^1$H-$^{15}$N-correlation NMR spectrum (2D $^1$H-$^{15}$N BEST-TROSY) of *S. aureus* SH1000 PG fragments solution obtained by digestion with mutanolysin and further processed with Lss, LssCD and LytMCD, respectively. Glycine residues with free carboxyl group released through enzymatic hydrolysis were marked against pale grey background. Glycines at acceptor stem-peptide were marked against dark grey background. Lysine side-chain (L-Lys$^{SC}$) and penultimate alanine (D-Ala$^4$)[54] produce altered NMR signals in enzyme treated versus untreated samples due to digestion of pentaglycine crossbridge. **d** Relative values of the intensities of NMR signals corresponding to glycine residues with free carboxyl group released through enzymatic hydrolysis from muropeptides. The intensity value for each peak is presented in relative to the sum of all the peak intensities identified in this region. Source data are provided as a Source Data file. **e** Relative abundance of MS intensities matching muropeptide digestion products generated by LssCD and LytMCD, expressed in relative to sum of all the MS intensities of the monitored species. The threshold for the product detection was arbitrary set for 10%. Source data are provided as a Source Data file.

enzymes (Fig. 2e). We detected increase in the mass intensity readout corresponding to the G$_5$-stem-peptide with four glycine cross-bridges in the samples containing LssCD and LytMCD reaction products. This confirms that the major digestion site of both enzymes is between the fourth and fifth glycine (Fig. 2b), consistent with the NMR digestion results (Fig. 2d). In conclusion, combining solution NMR and MS allowed us to explicitly define bonds hydrolysed by LssCD and LytMCD.

## Substrate complexity impacts the activities of LssCD and LytMCD

To explore enzymatic reactions in the context of the intact polymerised *S. aureus* sacculus, $^{13}$C-$^{15}$N-labelled PG was treated with mature Lss or LssCD (Fig. 3a). Digestion products were further processed by mutanolysin, and muropeptides were analysed by 2D $^1$H,$^{15}$N-correlation spectra. Mature Lss digests sacculi and produces a multipeak spectrum indicative for muropeptide digestion products.

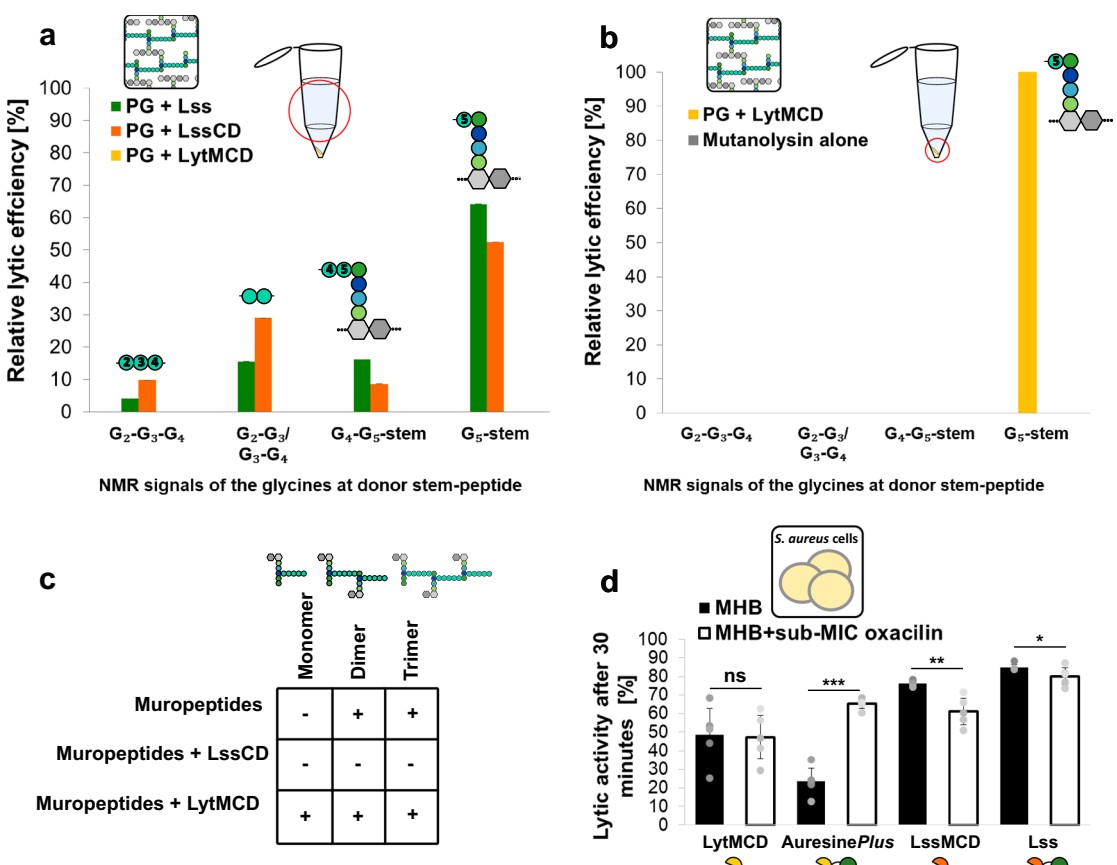

**Fig. 3 | Substrate complexity impacts the activities of LssCD and LytMCD.**
**a** Relative values of the intensities of NMR signals corresponding to glycine residues with free carboxyl group released in the supernatant through enzymatic hydrolysis from intact polymerised sacculus. The values were calculated as in Fig. 2d. Source data are provided as a Source Data file. **b** Relative values of the intensities of NMR signals corresponding to glycine residues with free carboxyl group released after (i) intact polymerised sacculus treatment with LytMCD and (ii) further treatment of insoluble material with mutanolysin. The values were calculated as in Fig. 2d. Source data are provided as a Source Data file. **c** Relative abundance of MS intensities matching muropeptide digestion substrates of LssCD and LytMCD, expressed in relative to sum of all the MS intensities of the monitored species. The threshold for the substrate detection was arbitrary set for 10%. Increased intensities for the masses of monomer detected in LytMCD sample are related to uncertainties of the

experiment. Source data are provided as a Source Data file. **d** Effect of diminished PG cross-linking on the lytic activity of the enzymes. Lytic activity of the enzymes against *S. aureus* TF5303 grown with or without sublethal antibiotic pre-treatment (decreased and regular PG cross-linking level; MHB+ sub-MIC oxacilin and MHB, respectively) monitored as turbidity reduction after 30 min. The results are presented in relative to the sample of bacteria suspended in the buffer without enzyme addition. Statistical analysis was performed using Student's *t* test for two independent means, ***$p < 0.0001$, **$p < 0.001$, *$p < 0.05$. $n = 5$ biologically independent samples. The expected *p*-values are 0.449102 for LytMCD, <0.00001 for Auresine*Plus*, 0.001515 for LssCD and 0.047524 for Lss. Data are presented as mean values +/− standard deviation. Abbreviations: MHB Mueller Hinton Broth. Source data are provided as a Source Data file.

However, no product of the reaction could be detected using only the Lss catalytic domain. This is in clear contrast to PG processing, completed by mature Lss (Fig. 3a). In an additional experiment, we increased the LssCD concentration 50 times to discriminate between the potentially low activity of the catalytic domain alone and its inability to cleave the substrate. Then, we observed in a NMR spectrum the peaks pattern corresponding to the digested pentaglycine bridge, which is indicative of PG sacculi solubilisation by hydrolase (Fig. 3a). This is also supporting the importance of the CBD of Lss ($CBD_{Lss}$) to enhance the lytic activity of LssCD on complex PG sacculi.

In contrast, under our experimental conditions, no reaction products of sacculus hydrolysis could be detected after treatment with LytMCD, even after the enzyme concentration was increased (Fig. 3a). In our experimental setup, only the supernatant of the reaction, i.e., soluble PG fragments released by the enzyme, was analysed. We then hypothesised that LytMCD could cleave the cross-bridge without releasing soluble fragments. To this end, insoluble PG obtained after LytMCD digestion was processed with mutanolysin and analysed by NMR (Fig. 3b). Indeed, we observed the signature of cross-bridge

cleavage between fourth and fifth glycine, which was absent in a control reaction with mutanolysin alone. This indicates that LytMCD can digest PG cross-bridges; however, in contrast to LssCD, soluble fragments cannot be released due to the extent of digestion or distribution of digestions into the sacculi.

More than 90% of the *S. aureus* muropeptides are involved to cross-linking of PG, meaning that most of them are grouped into highly polymerised multimers[40]. NMR digestion experiments provided a local view on the activity of the enzymes and enabled us to define the peptide bonds hydrolysed with high accuracy. To complement the knowledge with a global perspective on the lytic preferences of LssCD and LytMCD towards multimeric muropeptides, we evaluated the residual relative amount of the muropeptides identified by UPLC-MS that correspond to selected potential substrates of the enzymes, including uncross-linked muropeptide monomer and cross-linked muropeptide dimer and trimer (Supplementary Fig. 2a, Supplementary Table 2)[7,41]. Neither of the enzymes digest muropeptide monomers, and only LssCD digests the muropeptide dimers and trimers (Fig. 3c). This indicates that LssCD digests multimeric muropeptide forms, whereas LytMCD does not.

Building on this, we hypothesised that the level of PG crosslinking could affect the lytic preferences of these two enzymes. To verify this hypothesis, we cultured *S. aureus* in medium containing subinhibitory concentrations of oxacillin[42] (Supplementary Fig. 3), the β-lactam antibiotic, which interferes with cross-bridge formation leading to reduction in the highly cross-linked oligomeric muropeptides content in the PG[43–45]. *S. aureus* cultured in the regular medium and in the presence of subinhibitory concentration of oxacilin was used to evaluate the lytic activity of the enzymes in a turbidity reduction assay (Fig. 3d, Supplementary Fig. 4a). We observed a faster drop in optical density in the sample containing LssCD and bacteria with regular crosslinking compared to that of the sample containing LssCD and bacteria with reduced crosslinking. We performed an analogous assay using the mature version of this enzyme, which strongly binds CW through its binding domain, and found that mature Lss displays similar clearance efficiencies for strongly cross-linked PG and the PG with lower reticulation.

The lytic activity of LytMCD towards cells with regular and reduced crosslinking was also unaltered. Again, to validate this effect upon strong CW binding, we utilised a chimeric enzyme, which contains LytMCD fused to the binding domain of mature Lss, termed Auresine*Plus*[27]. In this case, the lytic activity of the enzyme towards cells with reduced crosslinking increased significantly. Cell lysis due to activity of autolysins was monitored and comprised a minor fraction of monitored lysis (up to 1% of all cells), indicative for that the observed cell lysis derives from the activity of externally added enzymes (Supplementary Fig. 4b). Based on this, we concluded that the level of PG crosslinking affects the lytic activity of LssCD and LytMCD differently.

## LytMCD and LssCD interact with muropeptides

The predominant cleavage site of LssCD and LytMCD is between the fourth and fifth glycines of the muropeptide (Fig. 2b). However, the presence of additional reaction products (multiglycine peptide for LytMCD and $G_4$-$G_5$-stem for LssCD) and distinct digestion patterns of the multimeric muropeptides obtained by MS analysis challenge the assumption that these enzymes recognise muropeptides in the same way. To investigate this further, their interaction with PG fragments was studied by solution NMR. To preclude ligand digestion, PG isolated from the *S. aureus* TF5311 mutant, which displays enriched serine-containing cross-bridges resistant to LssCD and LytMCD cleavage, was used[46]. To produce muropeptides, PG was digested overnight with mutanolysin, producing substrate analogues (Figs. 2a and 4a). In addition, PG of *S. aureus* SH1000 was digested with mature Lss to obtain M23 peptidase reaction products. This ligand (product) comprises the intact glycan chain (6 moieties on average)[1], stem peptide and disrupted glycine cross-link.

A series of 2D $^1$H-$^{15}$N correlation NMR experiments were recorded after the concentration of each ligand was increased. By titrating each protein with the ligands, we followed chemical shift perturbations (CSPs) of amino acid resonances, which are characteristic of their interaction between the ligand and the protein (Fig. 4a). The CSPs induced by the substrate analogue are of higher amplitude compared to those measured with the product, suggesting a lower affinity for PG fragments with disrupted cross-bridges (Supplementary Fig. 5a). The evolution of CSPs with ligand concentration enabled the estimation of dissociation constants of ca. 80 μM for both LssCD and LytMCD with substrate analogue, but only of 850 and 380 μM for product with LssCD and LytMCD, respectively.

The CSPs were then mapped on the enzyme surfaces (Fig. 4b, Supplementary Fig. 5b). Upon titration with the substrate analogue, the residues forming/in proximity to the binding groove of LssCD, namely, Y280, M327 and H328, were particularly affected (Supplementary Fig. 6). In contrast, the reaction product does not perturb those residues, indicating that they are only involved in the interaction

with an intact cross-bridge. This analysis could not be conducted on LytMCD due to incomplete NMR assignment.

The binding groove of M23 peptidases is formed by four flexible loops, and their architecture determines the specificity of the enzymes from this family[19,24]. Upon titration with both substrate analogue and product, the loops were involved in the interactions, indicating that they stay in close contact with the PG (Fig. 4c, Supplementary Fig. 5c). For LssCD, the residues forming loops 1 and 3 were perturbed irrespective of the ligand used (Supplementary Table 3). This indicates that they accommodate the stem-peptide and/or the glycan strand, rather than the cross-bridge, that is present only in the substrate analogue. Loop 4, on the other hand, is only affected by the interaction with the reaction product, suggesting that the loop is positioned close to the long glycan strand. For LytMCD, loop 3 is involved in the interaction with both the substrate analogue and product. Loop 2 and its neighbouring region were perturbed by titration with a product alone, which could be indicative of its interaction with an intact glycan chain.

Finally, we performed the HADDOCK docking protocol[39] to correlate our observations with a molecular model. The staphylococcal muropeptide monomer was used as the ligand (Supplementary Fig. 7a), and the solution state NMR data from the interaction of each enzyme with the substrate analogue were used to drive the docking (Fig. 4d, Supplementary Dataset 1 and 2). We analysed the four best energy models for the molecular contacts formed between protein side chains and the ligand using the Ligplot+ program[47] (Supplementary Figs. 7b, 8 and 9). In both models, the ligand displays the same polarity with respect to the binding cavity of the enzymes (Supplementary Fig. 7c). Loop 1 and β-sheet residues stabilise pentaglycine positioning at the bottom of the binding groove through multiple H-bonding/van der Waals interactions, in agreement with previous works on M23 peptidases[15,19]. $Gly_4$ is stabilised through an H-bond with LssCD H328, the catalytic residue conserved across M23 peptidases, and LytMCD Y204, which plays a role in the lytic activity and binding of LytMCD as described previously[15] (Supplementary Table 4). In both models, $D-Ala_4$-$D-Ala_5$ forms multiple molecular contacts with the residues localised in the corresponding regions of two enzymes at the entrance of the binding groove (LssCD K262, LytMCD K196 and Q199). $D-Ala_6$ at the donor stem-peptide is stabilised through hydrogen bonding between the loop 1 backbone and the H-bond formed with the LssCD Q363 side chain. Except for the LssCD Y307 backbone interaction, almost no contacts were established between the enzymes and the glycan chain. To summarise, both enzymes form a rich network of molecular contacts with the peptide part of the muropeptide, involving both cross-bridges and stem peptides.

## LytMCD and LssCD interact with the intact PG sacculus

PG hydrolases operate in the complex environment of the bacterial cell wall. If NMR interaction studies combined with docking also provided a molecular view of enzymes as they interact with isolated muropeptides, the context of heterogeneous intact PG, a much more complex ligand, could be resolved. Not many structural biology approaches are suitable to study heterogeneous insoluble ligands as intact PG, but it is accessible to magic-angle-spinning (MAS) solid-state NMR (ssNMR). Through this approach, $^2$H,$^{13}$C,$^{15}$N-labelled enzymes bound to insoluble unlabelled PG sacculus can be analysed. LssCD and LytMCD bind prominently to the sacculus purified from both *S. aureus* wild-type and TF5311 mutant, M23 peptidase resistant (Fig. 5a, b, Supplementary Fig. 10), so the latter served to extract PG sacculus and pull down the enzymes for the ssNMR study. The decrease in absorbance at 280 nm measured in a sample containing proteins mixed with sacculus after 1 h was indicative of their binding, and we estimated that 0.6–1 mg of each enzyme cosedimented with 3 mg of PG. No further drop in the absorbance of the supernatant at 280 nm was detected over a prolonged time (four hours), indicative of PG saturation.

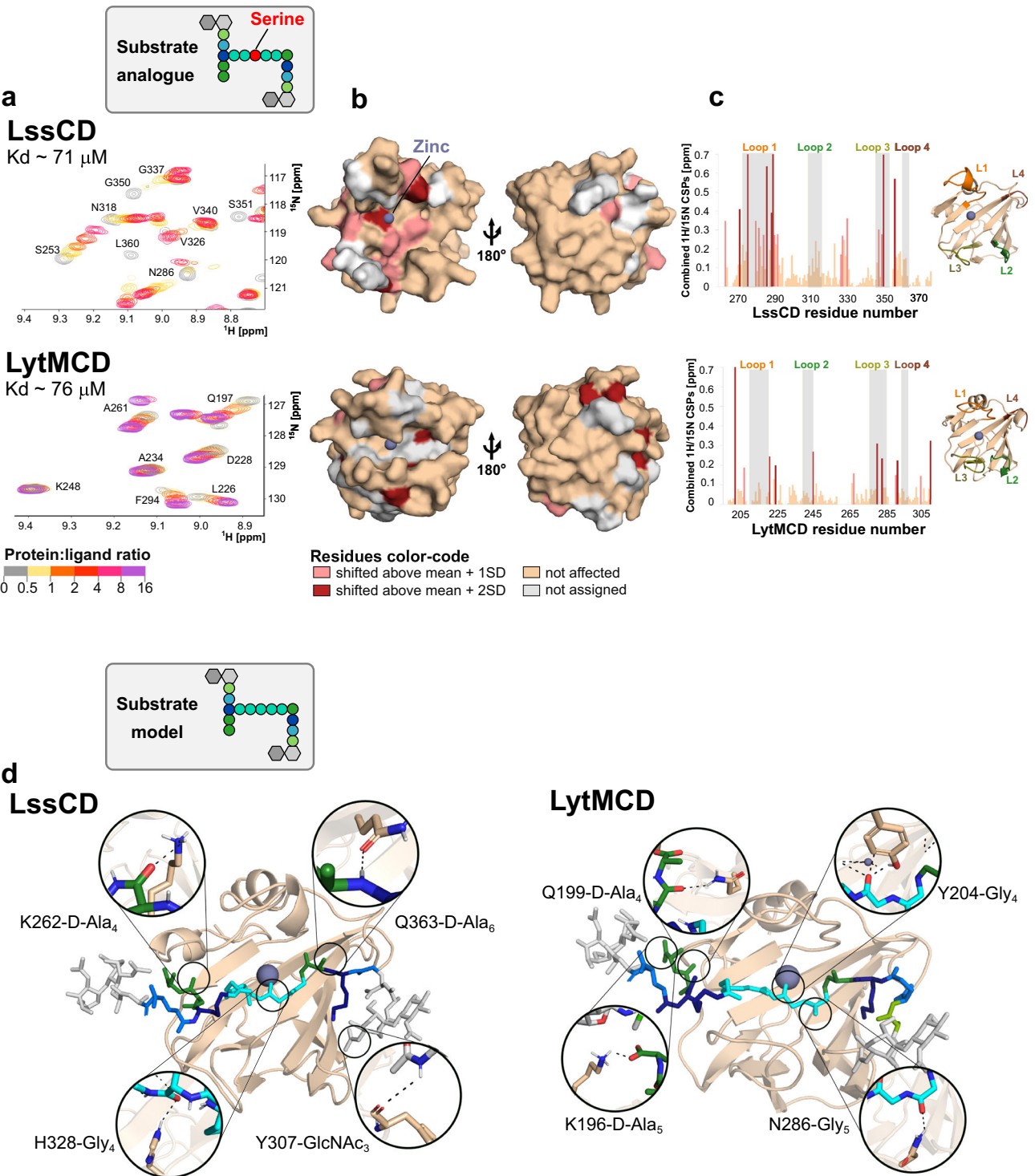

**Fig. 4 | LytMCD and LssCD interact with the muropeptide. a** Region of the ${}^1$H,${}^{15}$N-BEST-TROSY spectra of ${}^{15}$N-labelled LssCD and LytMCD recorded at 25 °C and pH 7.0. The experiment was conducted before (grey) and after addition of unlabelled substrate analogue at increasing ratios (colour scale from pale yellow to violet). The presented structures correspond to the Protein Data Bank (PDB) deposits (4ZYB for LyMCD and 5NMY for LssCD). **b** CSP superior to 1 standard deviation (SD > 0.25 ppm for LssCD; >0.13 ppm for LytMCD coloured in salmon) or 2 standard deviations (>0.38 ppm for LssCD; >0.20 ppm for LytMCD coloured in crimson) are displayed on the surface representation of the respective proteins. Unaffected and unassigned protein regions were coloured wheat and white, respectively. The left panel corresponds to the same orientation as the right one but rotated by 180°. The zinc is depicted as a blue-grey sphere. **c** Combined ${}^1$H and ${}^{15}$N chemical shift perturbations calculated for each residue from the data displayed in panel **a** as the weighted-average distance between the resonance position of the free form of LytMCD or LssCD and its equivalent position in the complex with substrate analogue measured at the ratio of 1:20 and 1:8, respectively. Bars are colour-coded with CSP amplitudes as in panel **b**. The loop regions are highlighted in grey. Loops are colour-coded on the chart and structure presented in the cartoon representation. Abbreviations: L loop. Source data are provided as a Source Data file. **d** Top-score structure of the best convergence cluster obtained for LssCD (PDB ID: 5NMY) and LytMCD (PDB ID: 4ZYB) HADDOCK models in complex with the substrate. Residues predicted to form hydrogen bonds with the ligand identified in at least 50% of the structures from the cluster of the best energy score are presented as sticks. Ligand is coloured as in the scheme in Fig. 1a. Muropeptide residues are numbered following the nomenclature provided in Supplementary Fig. S7.

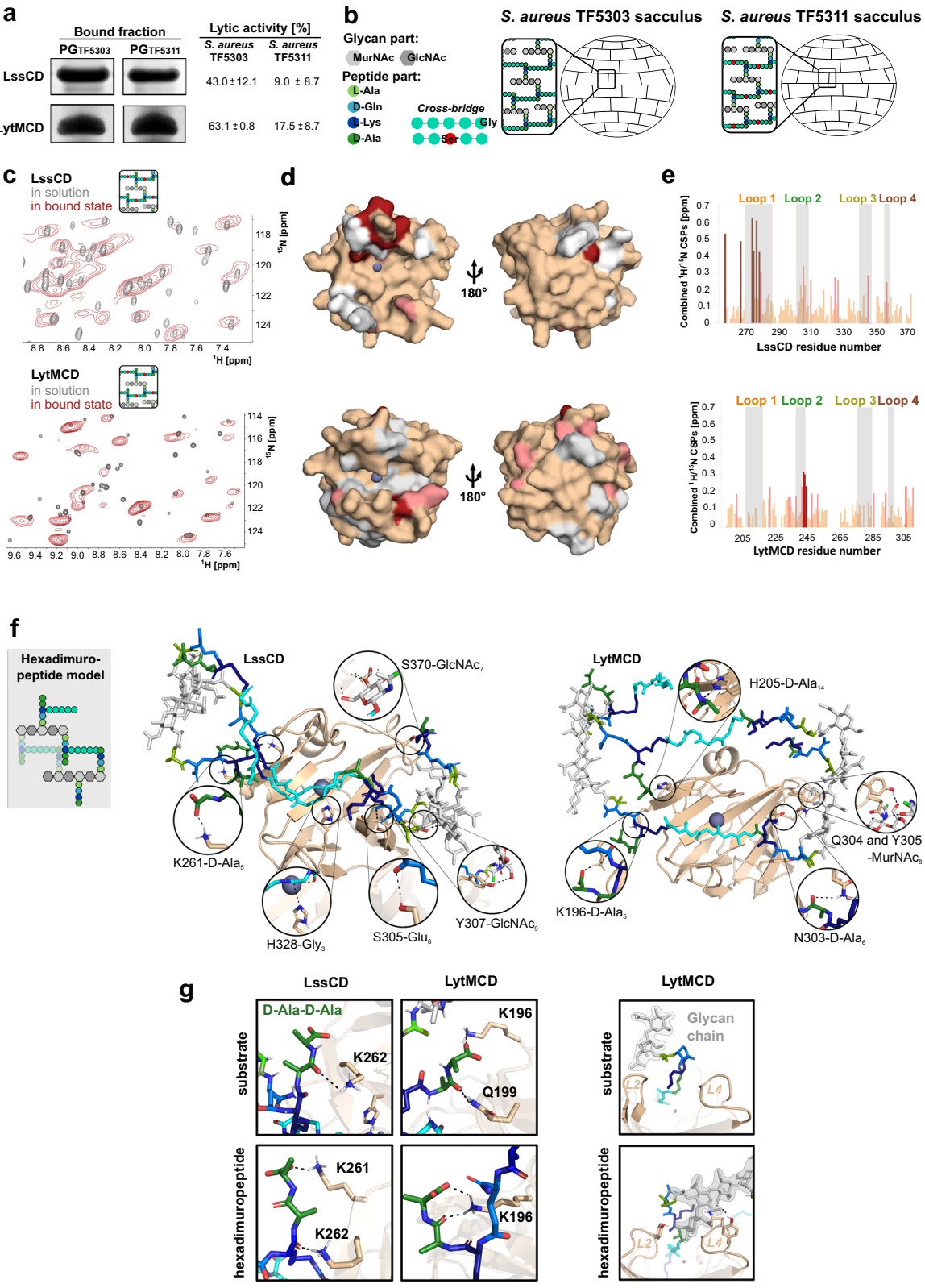

Samples containing PG-bound enzymes were packed into 1.3 mm NMR rotors, and $^1$H-detected 1D hcH NMR experiments were recorded. We first studied binding dynamics using either (1) cross-polarisation (CP) or (2) scalar-coupling-based refocused-INEPT for $^1$H to $^{13}$C (or vice versa) transfers prior to $^1$H detection. The former approach generates a signal derived from proteins bound to the sacculus, while the latter provides signals from highly flexible fragments of bound protein or from freely tumbling proteins. For both catalytic domains, no *versus*

strong signals were observed in INEPT and CP experiments, respectively, indicating that both LssCD and LytMCD tightly bind to the sacculus with long residential times ($k_{off} < \mu s$).

Then, we compared the $^1$H,$^{15}$N-correlation spectra of the free and PG-bound enzymes (Fig. 5c) recorded in the same buffer and temperature conditions in solution- and solid-state NMR, respectively. The overall similarities between the two spectra indicate that protein folding is preserved upon PG binding. To unambiguously identify

**Fig. 5 | LytMCD and LssCD interact with the PG sacculus. a** Samples containing LssCD and LytMCD bound to the intact, polymerised sacculus visualised by SDS-PAGE electrophoresis. Lytic activity of enzymes was evaluated in turbidity reduction assay against *S. aureus* TF5303 wild-type and TF5311 mutant during 1 h. The experiment was repeated three independent times and molecular weight marker is provided in fig. S10. **b** Schematic representation of the PG sacculus used for the protein pull-down assays. **c** Comparison of $^1H$, $^{15}N$-correlation spectra of uniformly $^2H$, $^{13}C$, $^{15}N$-labelled LssCD and LytMCD free in solution (grey) and in the presence of intact, polymerised PG (crimson) from *S. aureus* TF5311. **d** CSP mapping on protein structure of Lss (*upper panel*) and LytM (*lower panel*) catalytic domains. CSPs superior to 1 standard deviation (SD > 0.23 ppm for LssCD; >0.16 ppm for LytMCD coloured in salmon) or 2 standard deviations (>0.35 ppm for LssCD; >0.23 ppm for LytMCD coloured in crimson) are displayed on the surface representation of

respective proteins. Zn(II) is depicted as blue-grey sphere. **e** Combined H and N chemical shift perturbations calculated for each residue from the data displayed in panel **c** as the weighted-average distance between the resonance position of the free form of LytMCD or LssCD and its equivalent position in the form bound to intact sacculus. Bars are colour-coded with CSPs amplitudes as in panel **d**. The loop regions are highlighted in grey. Source data are provided as a Source Data file. **f** LytMCD or LssCD:hexadimuropeptide best energy model from the best HADDOCK cluster of solutions. Residues predicted to form hydrogen bonds with the ligand identified in at least 50% of the structures from the cluster of the best energy score are presented as sticks. **g** Comparison of HADDOCK models for the residues interacting with D-Ala-D-Ala (green colour) and glycan chain (grey colour) regions. For clarity, the glycan chain region was presented for LytMCD models only. Abbreviations: L loop.

amide resonances of the enzymes in the PG-bound state, 3D ssNMR backbone experiments were recorded and compared to the solution-state NMR backbone resonance assignments. Combined $^1H$,$^{15}N$-CSPs between free and PG-bound states were then calculated and reported on the protein structures. Both LssCD and LytMCD CSPs localised mostly around the binding groove (Fig. 5d). For LssCD, loop 1 experiences the most pronounced CSPs among all loops (Fig. 5e, Supplementary Table 3), in strong agreement with the interaction results on the soluble substrate analogue. Similarly, loop 2 of LytMCD is highly perturbed by the interaction with sacculus, soluble substrate analogue and product. This finding outlines the important role of LssCD loop 1 and LytMCD loop 2 in intact PG recognition.

Significant differences were observed between the interaction of the enzymes with the soluble PG fragments and sacculus (Supplementary Table 3). LssCD loops 2-4 are much less affected by sacculus binding than by soluble PG fragments. None of the residues forming LytMCD loop 3 are involved in the interaction with the sacculus, which clearly contrasts with the soluble PG fragments. This indicates that both enzymes form different molecular contacts with the complex sacculus and the short PG fragments.

Previous NMR studies on lysostaphin revealed that loop 1 is partially restrained and displays a transient α-helix that has not been identified in any crystallographic structure[30]. To determine if this region is prone to secondary structure rearrangements upon PG binding, we mapped $^{13}C\alpha$ CSPs that are more sensitive to local conformational changes than $^1H$ CSPs. Indeed, we observed distortion in the loop 1 region of the protein in the bound state (Supplementary Fig. 11).

Both catalytic domains form contacts with the sacculi. Similar to muropeptides, HADDOCK data-driven docking was performed using ssNMR CSPs and a more complex ligand, a hexadimuropeptide to simulate the PG (Fig. 5f, Supplementary Fig. 12, Supplementary Dataset 3 and 4). Models suggest that LssCD binds PG in the plane of the glycan strand, while LytMCD is tilted away from it by approximately 40°. Consequently, loop 1 of each enzyme is positioned differently (Supplementary Fig. 13). LytMCD loop 1 is ~5 Å from the cross-bridge not docked in the binding groove and in H-bond distance to the stem-peptide (H205 contacts D-Ala$_{14}$, Fig. 5f). In contrast, LssCD loop 1 is stretched ~15 Å above the neighbouring cross-bridge and does not form any specific contacts with this region of the ligand.

Multiple H-bonds or van der Waals interactions were formed with the ligand region docked in the binding groove of the enzymes (Supplementary Table 4, Supplementary Figs. 14 and 15). The docked cross-bridge displays the same polarity with respect to the active centre of each enzyme and is the same as that found in the substrate modelling. LssCD H328 and LytMCD Y204 form H-bonds with cross-bridge glycine residues consistent with substrate modelling. An additional H-bond was formed between LytMCD N286 and Gly$_5$, which was not previously identified. Stem-peptide D-Ala$_4$-D-Ala$_5$ is stabilised through H-bonds formed between lysine and glutamine residues, which were also identified in the substrate models (Fig. 5g). This suggests a possible

general mechanism involving the stabilisation of D-Ala-D-Ala of the terminal acceptor stem peptide by the polar residues grouped at the entrance to the binding groove. In addition, H-bonds were established between the glycan strand and loops 2 and 4 (Fig. 5g). LssCD Y307 and LytMCD Y239, the corresponding residues in loop 2, both form H-bonds with GlcNAc (ninth and seventh, respectively; for numbering see Supplementary Fig. 12). None of these contacts were identified by modelling with the substrate, indicating that increased ligand complexity induces a higher number of molecular contacts with the enzymes and glycan strand.

Given the low resolution nature of CSPs mapping and therefore certain degree of ambiguity of calculated structural models, we evaluated the importance of these interactions for the lytic activity of the enzymes, we performed site-directed (SD) mutagenesis (Supplementary Fig. 16). We observed that LssCD K261A displayed lytic activity similar to wild-type, and mutagenesis of the corresponding LytMCD K196 resulted in insoluble protein. The LytMCD Q199A variant displayed reduced lytic activity compared to that of the wild-type (by 65% after 10 min of reaction). Upon mutagenesis of tyrosine residues predicted to contact GlcNAc, the obtained protein variants displayed reduced lytic activity compared to that of the wild-type variants (LssCD Y307A by 17% and LytMCD Y239A by 80%). Through that, we validated the role of the interaction with D-Ala-D-Ala (LytMCD Q199) and the glycan strand (LytMCD Y239, LssCD Y307) for the correct positioning of the muropeptide in the binding groove, which subsequently affects cross-bridge hydrolysis.

## Discussion

LssCD and LytMCD share many common features, including similar folds (RMSD = 0.786), high sequence similarity (63%), high lytic activity towards *S. aureus*, and specificity towards the glycylglycine bond in PG[17,48]. We found that these enzymes hydrolyse the peptide bond between the fourth and fifth glycine of the muropeptide cross-bridge. HADDOCK models using NMR titration data revealed similar models of muropeptide docking. We concluded that LssCD and LytMCD interact with isolated muropeptides in a similar manner.

Differences between the enzymes emerge as the complexity of the ligand increases. First, we found that multimeric forms of muropeptides are digested efficiently by LssCD but not by LytMCD (Fig. 3c). Second, although Lss/LssCD and LytMCD process PG sacculi, only Lss/LssCD releases soluble PG fragments. Finally, we observed that LssCD and LytMCD bind PG sacculi differently. Hexadimuropeptide models revealed different angles at which enzymes approach the ligand (Fig. 6a) and consequently experience different molecular contacts with PG (Fig. 6b). We observed that enzymes form a richer network of molecular contacts with PG than with muropeptide. Considering discrepancies in the amount and type of digestion products, we concluded that ligand complexity differentiates the specificity of the enzymes.

Which PG features could play a role in that? Considering the low activity of LytMCD towards muropeptide multimers and PG sacculi, we

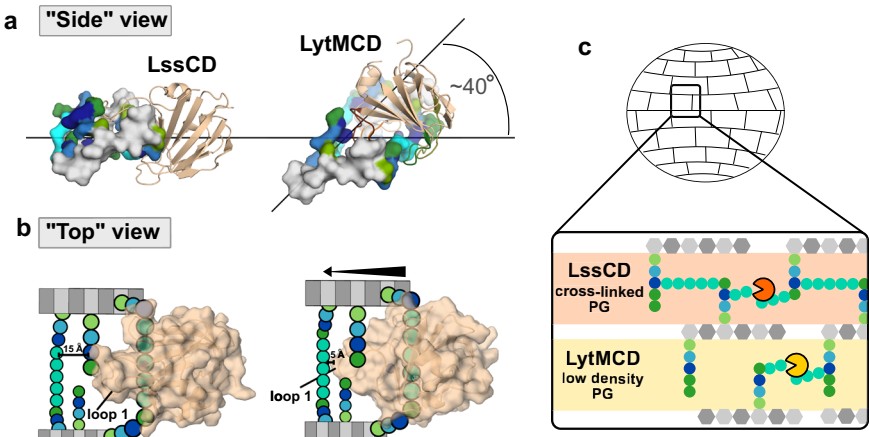

**Fig. 6 | Cross-linking affect the activity of Lss and LytM differently. a** HADDOCK models oriented with respect to glycan strand. Ligand surface is coloured as in Fig. 1. Enzymes are shown in cartoon representation. **b** Schematic representation of LssCD and LytMCD interaction with double cross-linked muropeptide. Arrows indicate the shift of LytMCD towards the neighbouring cross-bridge. Distance between loop 1 and glycine of the cross-bridge was estimated in Pymol. PG residues were colour-coded as in Fig. 1. **c** Proposition of a model for the differentiated LssCD and LytMCD activity within the PG mesh. LssCD solubilises cross-linked PG, whereas LytMCD targets preferentially sites of sacculus that display decreased cross-linking.

hypothesised that PG cross-linking could be this factor. Indeed, LytMCD (in the form of CW attached Auresine*Plus*) lysed *S. aureus* cells with decreased cross-linking more efficiently than *S. aureus* with regular cross-linking, and an inverse tendency was found for LssCD. Binding NMR-data driven model indicates that LytM loop 1 is positioned in proximity to neighbouring cross-bridge and forms molecular contacts with uncrosslinked stem-peptides, what is not a case for Lss. Based on that, we hypothesised that LytMCD activity is restrained through molecular contacts formed with polymerised PG, which serve as the regulatory mechanism to harness its lytic activity to certain spatial/temporal regions of the sacculus. If this reasoning is correct, LytM would be more active towards PG displaying decreased density of reticulation due to e.g. activity of other autolysins or stretching tension posed on cell wall during its expansion[49] (Fig. 6c). LytM is produced by *S. aureus* in high abundance[50], which underlies importance of its control on posttranslational level, and here we propose that architecture of its substrate contributes to this regulation. Regulation through PG crosslinking was recently described for another PG hydrolase of *S. aureus*, LytH, in which the lytic activity is limited to uncrosslinked sacculus regions[51]. In contrast, Lss is a highly effective enzyme, which evolved as a potent antimicrobial warhead. We observed that it is active towards polymerised PG and in our model forms less molecular contacts with polymerised PG, which in the proposed scenario alleviates substrate constraints and thus contribute to increased processivity of the enzyme. Therefore, we concluded that the limited number of molecular contacts formed between Lss and PG makes this enzyme to act in location and cycle-independent manner.

Despite rich literature on PG hydrolases[5,9-11], knowledge on their interaction with a natural substrate at the atomic level is clearly missing. M23 peptidases have been thoroughly characterised structurally[13,15,19] however, very little is known about their interaction with muropeptides/PG sacculi[15]. With this work, we fill this long-standing gap and provide structural details of the LssCD and LytMCD interaction with stem peptides and glycan strands. We identified polar amino acids grouped at the entrance to the binding groove, which stabilise the terminal D-Ala-D-Ala of the stem-peptide, and a rich network of H-bonds formed between loop 2 and 4 residues and the glycan strand. We evaluated their role by SD mutagenesis, which allowed us to identify residues that contribute to correct substrate positioning prior to catalysis.

Through solution- and solid-state NMR, MS, data-driven modelling and biochemical approaches, we provided mechanistic model of the enzymes operating in a complex mesh of the bacterial cell wall. To further develop the model of specific substrate recognition by hydrolases, it would be beneficial to extend the study with other molecular and biophysical techniques. Nevertheless, structural studies on protein-PG complexes are very scarce[36,] and no data-driven modelling with complex PG ligands, such as those presented here, has been published thus far. With this, we intended to provide a better understanding of the action and regulation of PG hydrolyses in other microbes and underpin future research efforts concerning the development of antimicrobial agents.

## Methods

### Cloning, production and purification of proteins

All the primers and constructs used in this study are listed in Supplementary Table 5. SD mutagenesis was performed using primers containing the desired mutations, and PCR amplification was performed using Phusion™ Plus DNA Polymerase (ThermoFisher Scientific, USA). The presence of the mutations was confirmed by sequencing. All the proteins used in the study were produced in *E. coli* BL21-Gold in M9 minimal medium. For $^{13}$C/$^{15}$N protein labelling, the medium was supplemented with $^{15}$N-labelled ammonium chloride and uniformly $^{13}$C-labelled glucose (Cambridge Isotope Laboratories, USA). For deuterated protein M9, 100% $D_2O$ medium and deuterated glucose were used. Protein production was induced by adding isopropyl β-D-1-thiogalactopyranoside (1 mM final concentration) when the culture reached $OD_{600}$ ~ 0.6. Production was continued overnight (O/N) at 18 °C. To produce unlabelled proteins, lysogeny broth was used. Bacterial pellets containing overexpressed proteins were resuspended in buffer B (20 mM Tris-HCl, 1 M NaCl, 5% glycerol pH 7.0). Lysates were centrifuged and dialysed O/N against buffer A (20 mM Tris-HCl, 20 mM NaCl, 10% glycerol pH 7.0) at room temperature (RT). The next day, the soluble fraction was diluted with 20 mM Tris-HCl, 5% glycerol pH 7.0 to lower the NaCl concentration to 10 mM. Lysate cleared by centrifugation was loaded on a 20 ml column containing ion-exchange resin (WB40S, Bio-Works, Sweden), and a gradient was performed in 20 column volumes from A to B buffer. Selected fractions were concentrated in Amicon centrifugal filters with 3 kDa cutoff (Merck, Germany), and size-exclusion chromatography was run in 20 mM Tris-HCl, 200 mM NaCl, and 10% glycerol pH 7.0 on a Superdex75 10/300 GL column using an ÄKTA™ Purifier system (Cytiva, USA). Concentrated and pure protein samples were flash frozen and stored at −80 °C prior to further analysis.

## Bacterial growth

*S. aureus* TF5303 (wild-type) or TF5311 (mutant strain containing serine insertion in cross-bridges, Supplementary Table 5)[33] used for PG purification were cultured overnight at 37 °C in 50 ml of tryptic soy broth (TSB) medium. To generate *S. aureus* displaying decreased cross-linking, culture was performed in MHB medium supplemented with subinhibitory concentrations of oxacillin (β-lactam antibiotic, Pol-Aura, Poland), which was experimentally determined for *S. aureus* TF5303 to be 0.125 μg/ml. The next day, it was refreshed with a fresh portion of MHB medium supplemented with oxacillin and cultured until $OD_{600}$ ~ 0.6.

## PG purification

PG was purified with a published protocol[41]. Briefly, bacterial cell cultures were centrifuged and resuspended in lysis buffer (100 mM Tris-HCl, 0.25% SDS pH 6.8) and boiled for 20 min. Then, the pellets were washed with Milli-Q® water at least 3 times, resuspended in 100 mM Tris-HCl, 1 μM $MgCl_2$ pH 8.0, placed in an ultrasonic bath for 30 min at RT and incubated for 1 h at 37 °C in a shaker with viscolase (250 U, A&A Biotechnology, Poland) and further with trypsin (50 μg/ml final) for another hour under the same conditions. To remove cell wall teichoic acids, pellets were incubated with 1 M HCl for 4 h at 37 °C. After that, they were washed in Milli-Q® water until neutral pH.

For muropeptide analysis, single batch of PG extract was digested overnight with mutanolysin (Sigma# M9901, Merck, Germany) in 20 mM Tris-HCl, 150 mM NaCl pH 7.0 buffer, and the solubilised fraction containing muropeptides was divided into three equal aliquotes and subjected to further processing by LssCD or LytMCD (2 μM final) or stored in the same conditions in which enzymatic reaction occurred (overnight, 37 °C). The next day, samples were reduced with sodium borohydrate according to a known protocol, and the neutralised sample was subjected to UPLC–MS analysis[41].

## Liquid chromatography and mass spectrometry

The muropeptides were loaded in 10 μl volume onto Acquity UPLC Protein BEH C4 column, 300 Å, 1.7 um, 1 mm × 50 mm (Waters) and eluted in an acetonitrile gradient by applying phase A (0.1% formic acid in $H_2O$) and phase B (0.1% formic acid in acetonitrile) during 12 min run (100% A for 2 min, 0%–40% B for 6 min, 40%–90% B for 2 min and 99% A for 2 min). The flow rate was 100 μL min⁻¹.

The MS analysis was performed on Synapt G2 MS, time-of-flight mass spectrometer equipped with a nanoACQUITY UPLC system (Waters). Sodium Iodide was used as a reference material for mass spectrometer calibration. The MS was set to ESI positive (ESI+) mode with a scan range m/z 100–3000. The capillary voltage was 2.5 kV. Sampling cone was set to 35 and extraction cone to 2.0. Source temperature was set to 80 °C and desolvation temperature to 150 °C. The flow rate of the desolvation gas was 650 L/h. Two blank runs were injected between the runs. The MS data were analysed using MassLynx software (V4.1; Waters). Mass spectra were deconvoluted using the MaxEnt3 algorithm provided with the Masslynx software with parameters set for minimal molecular mass of 200 Da, maximal molecular mass of 8000 Da and maximal number of charges 5. The relative abundance of observed masses that matched to theoretical masses of *S. aureus* PG fragments were analysed (Supplementary Tables 1 and 2). Only dominant forms of selected muropeptides were tracked.

## PG digestion assays

For the Remazol blue release assay[52], the purified PG was stained in Remazol Brilliant Blue R (RBB, Sigma# R8001, Merck, Germany) dissolved in 250 mM NaOH to a final concentration of 20 mM. The PG sample was incubated with the dye for 6 h at 37 °C and further incubated for 12 h at 4 °C. Then, PG was washed thoroughly with water until the supernatant turned transparent. 1 μM hydrolase and remazol-dyed PG were mixed together and the reaction topped up to 400 μl with 50 mM glycine, 100 mM NaCl pH 8.0 buffer. Final $OD_{600}$ of PG was ~0.6. PG digestion was performed overnight at 37 °C. The reaction was quenched with an equal volume of 96% ethanol, the sample was centrifuged, and the absorbance of the supernatant was measured in a microplate reader at 595 nm (Bio-Rad Laboratories, Inc., USA).

¹³C-¹⁵N-labelled PG from *S. aureus* SH1000 model laboratory strain[53], produced and purified before[54], was used for NMR digestion experiments. PG at $OD_{600}$ ~ 1 was resuspended in 20 mM Tris-HCl pH 7.0, 150 mM NaCl buffer and digested with mutanolysin (30 U) for 8 h at 37 °C. The sample was centrifuged for 5 min at 16,000 g, and the supernatant was further processed with the selected enzyme (1 μM Lss, 1 μM or 50 μM LssCD, 50 μM LytMCD) overnight at 37 °C. The next day, the sample was centrifuged, and the supernatant was analysed by NMR. To further analyse the sacculus processed by LytMCD, the pellet was washed multiple times with the reaction buffer. Then, it was treated with mutanolysin ON at 37 °C. The next day, the soluble fraction was subjected to NMR analysis.

## Peptidoglycan pull-down

The procedure was based on a published protocol[15]. In brief, a sample containing 5 μl of *S. aureus* TF5303- or TF5311-purified PG ($OD_{600}$ ~ 20) and 10 mg of enzyme was topped up to 50 μl with optimised reaction buffer (50 mM glycine, pH 8.0). The samples were incubated on the bench at room temperature and mixed every 20 min for a total duration of 1 h. To stop the reaction, the sample was pelleted at 16,900 × g. Bound (pellet) and unbound (supernatant) fractions were loaded on an SDS–PAGE gel, and the protein content of each fraction was visualised through Coomassie Brilliant Blue staining. The absorbance at 280 nm of the supernatant fraction was also monitored. Unprocessed SDS-PAGE gel was provided in Supplementary Fig. S10.

## Turbidity reduction assay

Overnight cultures of *S. aureus* TF5303 (or TF5311) were prepared by inoculating a single colony with fresh TSB medium. On the next day, the growth medium was refreshed, and the culture was grown until $OD_{600}$ ~ 1. The culture was centrifuged at 16,900 × g, and bacterial pellets were resuspended in 50 mM glycine, pH 8.0 reaction buffer, supplemented with 100 mM NaCl for experiments with mature Lss and Auresine*Plus*, the chimeric enzyme composed of the LytMCD and LssCBD domains[27]. For the assays on *S. aureus* TF5303 with decreased cross-linking, the buffer was supplemented with subinhibitory concentrations of oxacillin (0.125 μg/ml). If not indicated otherwise, 100 nM hydrolase was used for the turbidity reduction assay conducted with published protocols[27]. The lytic reaction was conducted at room temperature, and the turbidity ($OD_{600}$) was monitored every 10 min for 1 h using a microplate reader (Bio-Rad Laboratories, Inc., USA). The experiment was repeated three times. Enzyme lytic activities are reported as a percentage of the measured turbidity reduction.

## NMR spectroscopy

All solution-state NMR experiments reported in this study were collected on Avance III HD Bruker spectrometers equipped with helium-cooled cryogenic solution-state ¹H, ²H, ¹³C, ¹⁵N-resonance probes operating at 600, 700 or 850 MHz ¹H NMR frequency. All solution- and solid-state NMR spectra were processed with Topspin (Bruker) and analysed with CcpNmr software (version 3.0.4.)[55].

Sequential backbone ¹H, ¹³C and ¹⁵N-NMR resonance assignments of LssCD and LytMCD in solution were performed using a set of HNCO, HN(CO)CA, HNCACB and HN(CO)CACB triple-resonance experiments in their BEST-TROSY version completed with a 2D ¹H,¹⁵N-BEST-TROSY correlation spectrum. These data were collected at 298 K on 970 μM LssCD and 670 μM LytMCD ¹³C, ¹⁵N-labelled protein samples in 20 mM Tris-HCl, 150 mM NaCl buffer at pH 7.0 with 10% $D_2O$[56].

To analyse the *S. aureus* SH1000 PG digestion products, 2D ¹H,¹⁵N-BEST-TROSY spectra were recorded on soluble ¹³C,¹⁵N-labelled PG

fragment samples in 20 mM Tris-HCl, 150 mM NaCl, pH 7 buffer at 298 K. The PG sample processed by Lss (mature) was used as a reference, and previously collected HNCO, HN(CO)CA, HNCACB and HN(CO)CACB experiments[54] were further processed to specifically assign the resonances of the digested pentaglycine cross-bridge with a particular emphasis on the resonances at 114–117 ppm $^{15}$N-chemical shift. To quantify the digestion products, the intensity of each peak was quantified and normalised to the sum of the intensities of all peaks in the 114–117 ppm $^{15}$N-chemical shift region.

For interaction studies with peptidoglycan soluble fragments, subsequent 2D $^{1}$H-$^{15}$N BEST-TROSY experiments were collected on 70 μM $^{15}$N-labelled LssCD or LytMCD protein samples in 20 mM Tris-HCl, 150 mM NaCl at pH 7.0 with 5% D$_2$O. The indirect dimension acquisition time was set to 50 ms for $^{15}$N, and the direct acquisition was set to 70 ms in $^{1}$H. The experiments were recorded for spectral width of 36 and 12 ppm in indirect and direct dimensions respectively with a number of scans set to 32. Each protein sample was titrated with a stock solution in the same buffer of unlabelled PG soluble fragments previously prepared by PG digestion, dialysis against pure water, and lyophilisation. The protein-to-ligand ratio for each titration point was quantified using integration of $^{1}$H resolved signals from the protein and from the anomeric carbohydrate signals of PG fragments. Chemical shift perturbations were calculated as a weighted average distance between the ligand-free protein sample and the protein sample at the considered ligand-to-protein ratio. Default weighting parameters from the CcpNmr software were used[55].

Solid-state NMR experiments were collected at a 950 MHz $^{1}$H Larmor frequency on a Bruker Avance NEO console using a 1.3 mm $^{1}$H, $^{2}$H, $^{13}$C,$^{15}$N CP-MAS probe. Samples resulting from the pelleted fractions of unlabelled PG pulldown experiments (see section above) with $^{2}$H, $^{13}$C,$^{15}$N-labelled protein were filled into 1.3 mm MAS rotors by ultracentrifugation (50,000 × g, in a Beckman SW32Ti rotor) for 1 h. All experiments were run at 55-kHz MAS and at a sample temperature of 298 K (external calibration). Cooling was achieved with a cooling gas flow at 260 K, while the bearing and drive gas flows were ca. 293 K. For PG-bound LssCD or LytMCD, 2D hNH and 3D hCONH, hCANH spectra[57] were recorded using proton-detected experiments entirely based on CP steps for the magnetisation transfers. In addition, a 2D hNH experiment involving INEPT transfers was also collected. Typical 90° pulse durations were 2.1 μs at 26 W for $^{1}$H, 3.8 μs at 23 W for $^{13}$C, and 4.7 μs at 60 W for $^{15}$N. The indirect dimension acquisition time for triple resonance experiments was typically set to 11.5 ms for $^{15}$N, 7.8 ms for $^{13}$CO and 5.1 ms for $^{13}$Cα. HN CP contact time was obtained with a transfer of 1 ms and with a radio-frequency field ramping from 6 to 10 kHz on $^{1}$H and 40 kHz on $^{15}$N. For HCA and HCO CP, a ramp on 1 H from 10 to 17 kHz was set for LytM sample and from 6 to 10 kHz for LssCD with 3–4 ms of a contact time. Cα and CO RF field were set to 40 kHz. For INEPT-based transfers, the delay was set to 2 ms for the hNH experiment.

## HADDOCK modelling

The docking of the protein on the peptidoglycan was performed using HADDOCK modelling protocols version 2.1 and 2.4[39]. As the input structures, we used the solution structure of LssCD (PDB ID: 5NMY) and the crystallographic structure of LytMCD (PDB ID: 4ZYB), which were processed with HADDOCK tools. The set of ten structures of a hexameric muropeptide fragment described before[36] or designed for this study hexadimuropeptide fragment served as docking ligands. Hexadimuropeptide was built as already described[36]. Briefly, the hexadimuropeptide structure was generated using Crystallography and NMR system software[58,59] version 1.3 while constraining interglycan strand distances to obtain parallel glycan strands. One hundred structures were refined in explicit water, and the best 5 structures were retained for docking. Those 5 structures were docked with Haddock 2.1 with either LssCD or LytMCD and 4000, 800 and 400 structures for

the first and second iterations and refinement in explicit water, respectively. Clustering was performed with HADDOCK tools and an 8 Å rmsd and 5 structures minimum per cluster. The best 5 structures of the cluster with the lowest HADDOCK total combined energy were retained for analysis.

The docking protocol exploited ambiguous restraints on protein residues, which showed chemical-shift perturbation in H−N experiments larger than twice the standard deviation over the whole sequence. For the muropeptide ligands, in which no unambiguously identified binding sites were available, all atoms were defined as passive ambiguous interaction restraints. The knowledge about the active site of M23 peptidases and major digestion products was exploited by adding an unambiguous distance restraint between the catalytic zinc ligand residues (H277, D288, H361 for LssCD and H210, D214, H293 for LytMCD) and the carboxylic groups of glycine four and five forming the cross-bridge of the muropeptide. HADDOCK restraints and energy statistics are summarised in Supplementary Table 6.

## Reporting summary

Further information on research design is available in the Nature Portfolio Reporting Summary linked to this article.

## Data availability

The NMR assignments for LytMCD and LssCD data generated in this study have been deposited in the BMRB database under accession codes 52145 and 52146, respectively. The mass spectrometry proteomics data have been deposited to the ProteomeXchange Consortium via the PRIDE[60] partner repository with the dataset identifiers PXD045859, PXD045856 and PXD045765. The HADDOCK models generated in this study are provided as Supplementary Data Set). PDB accession numbers of LssCD and LytMCD used in this study are 5NMY and 4ZYB. The data that support the findings of this study are available from the corresponding author upon reasonable request. The authors declare that all other data supporting the findings of this study are available within the paper and its supplementary information files. Source data are provided with this paper.

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

## Acknowledgements
Authors like to thank Paweł Mitkowski (MMRI, PAS), for preparing unlabelled LssCD variant and sharing chemical scheme of the muropeptide multimer in ChemSketch, Jacek Olędzki from MS Laboratory (IBB PAS) for running MS assays, Isabel Ayala (IBS) and Waldemar Vollmer (Newcastle University) for kindly sharing with us $^{13}$C, $^{15}$N labelled PG from *S. aureus* SH1000. The research was founded by the Polish National Agency for Academic Exchange NAWA: International Academic Partnerships as the part of the program on the Molecular Basis of Enzyme Specificity and Applications (PPI/APM/2018/1/00034) received by I.S., the iNEXT-Discovery project: Determination of the structural aspects of peptidoglycan hydrolases interactions with the bacterial cell wall (PID: 15212), which covered the cost of 28 measurement days at the National and European NMR platform located at IBS received by A.R., the Train2Target project granted from the European Union's Horizon 2020 research and innovation program under the Marie Sklodowska-Curie (grant agreement number 721484) received by J.-P.S. This work used the platforms of the Grenoble Instruct-ERIC center (ISBG; UAR 3518 CNRS-CEA-UGA-EMBL) within the Grenoble Partnership for Structural Biology (PSB), supported by FRISBI (ANR-10-INBS-0005-02) and GRAL, financed within the University Grenoble Alpes graduate school (Ecoles Universitaires de Recherche) CBH-EUR-GS (ANR-17-EURE-0003).

## Author contributions
A.R. performed the experimental studies, analysed data, prepared manuscript; C.L. performed HADDOCK analysis using modelled hexadimuropeptide ligand; A.V. run solid-state NMR experiments; C.B. collected and analysed muropeptide spectra and edited the manuscript; M.K.-D. validated MS analysis and described UPLC-MS methodology; I.S. initiated the project, participated in manuscript preparation, provided funding; J.-P.S. designed a study, participated in manuscript preparation, provided funding, supervised the work.

## Competing interests
The authors declare no competing interests.
