## [Peer Review File · Nature Communications]

REVIEWER COMMENTS

Reviewer #1 (Remarks to the Author):

The authors present the results of their studies concerning 2 hydrolases of the peptidoglycans backbone of *Staphylococcus aureus*.

The authors successfully produced meaningful 2D NMR data to elucidate the specificity of the Lysostaphin (Lss) and LytM enzymes in contact with soluble fragments of muropeptides and insoluble fractions assumed to mimic the behavior of the whole peptidoglycan wall. Modeling study supported well the NMR data and the author proposed the mechanistic of the enzyme-substrate binding and the residues implied in the specific cleavage position of the pentaglycine peptide bridges at different position, depending on the enzyme.

Atomic resolution of 3D structures in relatively complex medium is of very interest due to the analytical challenges and probably open the way to further similar enzymatic study in almost biological conditions.

The peptidoglycan is a major target for preventing bacterial infection. The appearance of multidrug-resistant bacteria (especially *S. aureus*) and the resistance against antibiotics limiting the formation of peptidoglycan is of interest. Improvement of the knowledge concerning the regulation mechanisms of peptidoglycan formation is a potential new weapon for clinical application.

The data generally support well the discussion, even if I can have some minor disagreement in some point as described later, especially the NMR data in combination with modeling by docking experiments.

Due to the lack of description concerning the MS data, there is no way for anybody to critically assess these data. The MS data "presented" by the author cannot be reproduce from the information mentioned in this paper (because there is none excepted one citation). The recent literature concerning the characterization and absolute quantification of soluble fractions of peptidoglycan by coupling mass spectrometry with chromatography or capillary electrophoresis was not covered in this paper by the authors to justify their approach.

More classical methods complete well the study and provide the controls required to support the hypotheses of the authors.

Despite the lack of description of the MS data, the work can still be published after revisions, especially about the MS data. The NMR data looks being solid rock and made me giving a favorable opinion for publication (once again after revisions).

General comments:

Some abbreviation are barely used. Some of them are used only once and can be removed to use the full expression instead. It will make the reading more fluent.

L117 :

The paper or the supporting information does not contain any material and method information concerning the liquid chromatography and mass spectrometry.

I invite the authors to provide these information, at least in the S.I, and a short mention about them in the plain text of the manuscript.

L119 :

The authors have to mention what were the m/z monitored to perform the semi quantitative data.

Due to the sample preparation of the peptidoglycan, an huge excess of NaBH₄ was used. This will generated sodium (and potassium) adducts of the detected species, additionally to the protonated species.

In the absence of an internal standard or (stable) heavy isotope labeling standard, all the combination of proton/sodium/potassium adducts have to be monitored for each charge state of the muropetide.

Moreover, in-source degradation of the muropetides such as anhydro-muropetide (neutral loss of water) and GlcNAc losses will be formed in source and have to be considered too.

Figure 2e: what does the dashed lines stand for ?

It seems to be related to the basal level of muropetides in the absence of the enzymes ?

L151:

This is meaningless without the description of the monitored m/z ions as specified in my previous comments for line 119

L159-160: Please rephrase. "Complex peak pattern" does not properly describe a 2D NMR data

L161: The authors are still referring to Lss ?

The reading of the 1st paragraph under the section "Substrate complexity impacts the activities of LssCD and LytMCD" are sometimes confusing and deserve some rephrasing for clarity

L161: "However, no product of the reaction could be detected using only the catalytic domain" I think the Fig. 2d referenced to this sentence and not the previous one in the text

L162: I think the end of the sentence is referring to the Fig. 3A that should be mentioned in the text "This is in clear contrast to mucopeptide processing, completed by both mature Lss and its catalytic domain alone"

L163: this sentence has no direct link with the previous one. Please rephrase it, e.g. by adding "In an additional experiment, we increased the LssCD concentration (...)"

L164: Please rephrase. "Multi peak detection" is meaningless because it is the purpose of the NMR: see the coupling of the nuclear spin due to the molecular orbital shielding...

You can rephrase as follow: "we observed in the NMR spectrum the peaks pattern corresponding to the digested pentaglycine bridge"

L166: change "demonstrated" by "this is also supporting".

L178: 10% of what ? Of the NMR signal ? Something else?

I propose that the authors rephrase the sentence as follow:

"S. aureus PG displays a high degree of cross-linking, as uncross-linked mucopeptides (monomers) account for less than 10%" to "More than 90% of the mucopeptides are involved to cross-linking of PG"

L184: sentence revision: there was a "the" missing

"(...) enzymes digest mucopeptide monomers, but only LssCD digests the mucopeptide dimers and trimers"

L187 – 194: Here is the real problem with this work concerning the LC-MS data:

In the total absence of Material & Method section concerning the HPLC and MS experiments, and in regards of the cited literature, one has to assume the HPLC method used by the authors was a reverse phase HPLC method, inspired from the paper of Kuhner and coworkers.

There is no description about the adaptation required to make it working in the context of coupling MS with HPLC.

If my initial guess is correct, in this case, there is no way that the monomer, dimer, trimer or tetramer of soluble muropeptide fragments to be eluted within less than 12 minutes.

This is clearly shown in the paper from Kuhner and coworkers in 2014 (DOI 10.1038/srep07494) and confirmed by Boulanger and coworkers in 2019 (DOI 10.1002/elps.20190047) for closely related muropeptides that the ones investigated by the authors.

In other word, all multimeric species detected below 12min using the same HPLC method from the work from Kuhner and coworkers correspond to chromatographic fractions that experienced carry-over between LC runs! This is obviously something we do not want, never!

In this regards, the authors have to fully describe their method and justify how/why the multimeric muropeptides can be eluted before 12 minutes, or revisit their data and the text before publication.

In my opinion, the only reliable LC-MS data is the absence of detection of some muropeptides (figure 3C) but the detection limit have to be assessed, or at least properly estimated or discussed. By chance the muropeptides were detected for the other experimental conditions, meaning that the limit of detection is not critical figure of interest, and that the relative comparison of the normalized LC-MS signals still support well the discussion of the authors.

Without replicates and estimation of the standard deviations, and in the absence of reliable and robust quantitation strategy, MS data from the figure 2 has to be interpreted with extreme caution.

L198-200: In regards to the Figure S4, my opinion is that the mature Lss enzyme had the same clearance efficiencies (in terms of turbidity reduction which correspond to the lytic activity) for the strongly cross-linked peptidoglycan (PG) and the PG with lower reticulation

L200: Consequently you can delete "In contrast" because LytMCD and mature Lss acted similarly whatever the level of reticulation of the PG, always according to the data of fig. S4

Figure 3

legend of Fig3b: I think it will be more clear and relevant to use “mutanolysine alone” instead of “no enzyme”

figure caption 3b: there is a typo, you wrote “resissues” instead of “residues”

y-axis of figure 3c: you should use “Residual relative amount of muuropeptides identified by HPLC-MS” instead of “relative intensity of the mass signal”. Also adapt the discussion in the plain text accordingly.

caption of 3c: did you mean “Total Ion Current (TIC) within the chromatographic peak” instead of “to the sum of all the intensities” ?

Fig3c: “The readout of muuropeptide dimers identified in the LytMCD processed samples is higher as compared to unprocessed sample, which could stem from processing of non-analysed muuropeptide multimers”

In the absence of replicates, it is more reasonable to assume that it is related to the uncertainties of the experiment, i.e. your standard deviation, especially in the absence of any kind of (internal) standard

It will reasonably be better to show normalized data in regard of the signal of “Muuropeptides” for each case (i.e. monomer, dimer, trimer) than a sum of signal which are meaningless in regard of the MS method, assumed to not use any (semi)quantitation with (internal) standards

Note that the remarks is also tue for the MS data in figure 2.

L231:

please use “by” instead of “in” in the sentence “ (...) distinct digestion patterns of the multimeric muuropeptides obtained BY MS analysis challenge the assumption that (...)”

L235: I think you meant Fig 2a and Fig 4a (which refer to the structure of the “substrate analogue” containing one serine instead of a glycine in the cross-linker peptide of (Gly)5

L258: Please use “suggesting” instead of “implying”. I think the data are not strong enough to affirm such statements about the loop, at least at this stage of the “Results” sectioning

L265: What “Ext. Data Fig 1 and 2” stand for ? Did you mean 424312_0_data_set_413380_rsqp07.txt and 424312_0_data_set_413381_rsqp84.txt ?

L267: I think you meant Fig. S9 instead of Fig. 9

L323: I am pretty sure that koff is a kinetics constant, consequently, the unit of koff are in min⁻¹ and not in concentration (μM in the manuscript),

kd would be in M⁻¹ and kon would be in M⁻¹.min⁻¹

By the way, even your sentence strongly support a scale time unit instead of concentration.

Please correct or rephrase

L348: What "Ext. Data Fig 1 and 2" stand for ? 424312_0_data_set_413382_rsqp9j.txt and 424312_0_data_set_413383_rsqqbp.txt ?

Additionally, I get some trouble with the visioning of the PDB structure where some of the 3D part of the protein were missing (PyMol v2.3.0 under Linux).

Honestly, I did not had the time to investigate the structure of these files and determine if it was a problem related to my version of PyMol or with the input files.

L451: Material and Methods

As mentioned above, there is absolutely not detail about the liquid chromatography and the mass spectrometry detection, neither in the supporting information.

The type of chromatographic pumps and column and the stationnary phase, dimension, mobile phases, gradient, flow rate, injection volume and condition, if any guard column or trap column was used ?

The mass spectrometer is also not described. There is a mention of a Time of Flight in the legends of some figures in S.I. Please describe properly the instrumentation, as well as the ionization mode (assumed to be electrospray positive ionization mode), source temperature and desolvation gas flow rate, the mass range, what was the average mass accuracy and/or the RMS of the mass accuracy within the mass range

What were the monitored m/z ? Is any standard was used to perform the (semi)quantitation ? Is the in-source fragmentation products were monitored together with the precursors ions ?

What was the versioning of MassLynx ? Did you smooth the peak? What was the integration algorithm and the settings ?

Is any blank injected between runs ?

I have to assume the author performed LC-MS coupling on-line, which can be quiete challenging without some good expertise for the LC-MS investigation of muropeptide. Nothing in the manuscript demonstrated or support such mastering and the reader need to have "blind and absolute" confidence.

Without these basic information, there is no possible evaluation of the LC-MS data

In the Material and method, there is some abbreviation that are never used. Please correct the text accordingly.

To the contrary, some of them are used but never described, e.g. WTA which I assumed to mean "Wall Techoic Acids"

"Cloning, production and purification of proteins"

L 496: What kind of Amicon did you used ? 1KD, 3KD, 10KD, 10KD ?

"PG purification"

The author barely justify the use of the strain SH1000. Please refer to your text or mention briefly the type of produced muropeptides that you used in your study as a control.

Authors have to motivate why the first 12 minutes of the LC runs are showed while there is almost no chance that all the muropeptides could be eluted from a reverse phase column even without the use of a pairing agent such as trifluoroacetic acid (or why not heptafluorobutyric acid).

L500: "The lytic assay on Remazol-dyed PG was performed overnight with 1 μ M hydrolase at 37 °C in 50 mM glycine, 100 mM NaCl pH 8.0 buffer against stained PG of OD600 ~ 6"

Please rewrite, this section is unclear. Moreover I think you meant OD~ 0,6 instead of 6

L515:

Did you mean enzyme instead of protein ?

L546:

Change "analysed" for "processed"

L549:

What is the "in this area" you mentioned ? Please rephrase and complete this sentence.

Reviewer #2 (Remarks to the Author):

In this manuscript, the authors conducted a comprehensive analysis of the enzymatic activity and specificity of two closely related M23 hydrolases, lysostaphin (Lss) and LytM, utilizing solution NMR techniques, Chemical Shift Perturbations (CSPs) of amino acid resonances, and magic-angle-spinning solid-state NMR (ssNMR). The findings revealed distinct binding patterns of the catalytic domains of these enzymes to the cell wall and differential effects of peptidoglycan cross-linking on their activity.

The manuscript offers a compelling strength as the authors successfully constructed a detailed model that maps the interaction between the M23 hydrolases and cell wall substrates at the amino acid level. Also, the manuscript is clearly written, and the provided data generally substantiate the proposed model. Nonetheless, I have identified a few concerns regarding this study, which are outlined below.

1. The physiological relevance of the findings in this study raises some questions. While the study primarily focused on the catalytic domains, Fig. 3 highlights the crucial role of the cell-wall binding domain (CBD) in enzyme activity. Consequently, it remains unclear to what extent these results can be extrapolated to intact cell wall hydrolases.
2. The binding model does not explain why the activities of Lss and LytM are affected by cell-wall cross-linking.
3. There appears to be a discrepancy between Fig. 3d and Fig. S4 (LssCD vs Lss) in the manuscript. While Fig. S4 indicates that Lss equally digested cells grown in both MHB and MHB+ sub-MIC oxacillin, Fig. 3d suggests that Lss more efficiently digests cells grown in MHB compared to those grown in MHB+ sub-MIC oxacillin.
4. The turbidity reduction assay depicted in Fig. 3b and d would benefit from additional clarification. It is puzzling that LytMCD failed to hydrolyze sacculi in Fig. 3b, yet it appears capable of lysing *S. aureus* in Fig. 3d. Considering that *S. aureus* is known to produce numerous cell wall hydrolases (at least 18), it is plausible that other hydrolases present on or within *S. aureus* cells might contribute to the observed cell lysis. To address this concern, it may be advisable for the authors to include a no-enzyme control in their experimental design, which would help elucidate the specific contribution of LytMCD in the context of cell lysis.

Reviewer #3 (Remarks to the Author):

The manuscript by Razew et al. describes an investigation of the binding and substrate specificity of two members of the M23 hydrolase family that cleave the penta-glycine crossbridge peptide in *S. aureus* peptidoglycan (PG). These include the *S. aureus* LytM protein that helps to remodel PG during growth and division, and the Lss enzyme that is secreted by *S. simulans* to eliminate bacterial competitors. Prior structural studies have deduced the mechanism of glycine peptide binding by this enzyme class. Here, the authors seek to gain insight into how these enzymes engage their substrates within the natural context of the bacterial cell wall. This is very challenging as PG is a high molecular weight and heterogeneous structure. Using mutanolysin degraded fragments of PG as enzyme substrates the authors convincingly localize the primary site of cleavage (between the 4th and 5th glycine residues). The authors also present data that reveals that the enzymes digest multimeric forms of PG differently, and that they bind to PG differently. Finally, HADDOCK derived models using NMR chemical mapping data suggest that they engage their substrates differently which may explain their distinct substrate preferences.

Strengths of this work include the application of a broad set of biophysical tools to investigate how biologically important PG hydrolases recognize their substrates in the natural context of the cell wall. The data seem to nicely support the idea that distinct structural features in the enzymes may alter the contacts each enzyme makes with PG components that surround the cross bridge peptide. However, there are major weaknesses. First, it is unclear if the molecular models presented in this paper are accurate to the level of detail that is claimed. In particular, figures 4D and 5F show specific enzyme-PG interactions that seem highly speculative given the low-resolution nature of chemical shift mapping data and the fact that the 3D structure of the PG can only be assumed. Thus, I am not convinced that they have determined at high resolution the details of how these enzymes engage PG sacculus (a central claim made in the abstract). Second, it is unclear if the mutanolysin degraded PG is an accurate surrogate for the intact cell wall that surrounds the microbe. Thus, making biologically relevant conclusions about how the enzymes function to degrade the cell wall seems speculative. Third, although the results are described in detail it is unclear why they are significant. Why is it important that the Lss and LytM enzymes engage and digest PG differently? Are the results predictive of enzyme function in other microbes?

We would like to thank all the reviewers for positive feedback, careful reading of the manuscript and exhaustive revision. We addressed all the raised comments point-by-point and introduced changes in the manuscript and supplementary material with tracked changes. In our view, this greatly improved the work in terms of clarity and logic of the presented story.

REVIEWER COMMENTS

Reviewer #1 (Remarks to the Author):

The authors present the results of their studies concerning 2 hydrolases of the peptidoglycans backbone of *Staphylococcus aureus*.

The authors successfully produced meaningful 2D NMR data to elucidate the specificity of the Lysostaphin (Lss) and LytM enzymes in contact with soluble fragments of muropeptides and insoluble fractions assumed to mimic the behavior of the whole peptidoglycan wall. Modeling study supported well the NMR data and the author proposed the mechanistic of the enzyme-substrate binding and the residues implied in the specific cleavage position of the pentaglycine peptide bridges at different position, depending on the enzyme.

Atomic resolution of 3D structures in relatively complex medium is of very interest due to the analytical challenges and probably open the way to further similar enzymatic study in almost biological conditions.

The peptidoglycan is a major target for preventing bacterial infection. The appearance of multidrug-resistant bacteria (especially *S. aureus*) and the resistance against antibiotics limiting the formation of peptidoglycan is of interest. Improvement of the knowledge concerning the regulation mechanisms of peptidoglycan formation is a potential new weapon for clinical application.

The data generally support well the discussion, even if I can have some minor disagreement in some point as described later, especially the NMR data in combination with modeling by docking experiments.

Due to the lack of description concerning the MS data, there is no way for anybody to critically assess these data. The MS data "presented" by the author cannot be reproduced from the information mentioned in this paper (because there is none excepted one citation). The recent literature concerning the characterization and absolute quantification of soluble fractions of peptidoglycan by coupling mass spectrometry with chromatography or capillary electrophoresis was not covered in this paper by the authors to justify their approach.

More classical methods complete well the study and provide the controls required to support the hypotheses of the authors.

Despite the lack of description of the MS data, the work can still be published after revisions, especially about the MS data. The NMR data looks being solid rock and made me giving a favorable opinion for publication (once again after revisions).

General comments:

Some abbreviations are barely used. Some of them are used only once and can be removed to use the full expression instead. It will make the reading more fluent.

This was inspected throughout the manuscript and corrected.

L117 :

The paper or the supporting information does not contain any material and method information concerning the liquid chromatography and mass spectrometry.

I invite the authors to provide these information, at least in the S.I, and a short mention about them in the plain text of the manuscript.

We agree with the reviewer that this information was not initially adequately presented . In the revised version we introduced the description for the liquid chromatography and mass spectrometry in Material and Methods and provided data in support for these analyses in SI (Fig. S2a, S3b, Tables S1-3). We updated MS figures in the main text as well (Fig. 2e, Fig. 3c). In addition, to validate our conclusions we implied a UPLC MS specialist whose name has been added in the list of authors.

L119 :

The authors have to mention what were the m/z monitored to perform the semi quantitative data.

Due to the sample preparation of the peptidoglycan, an huge excess of NaBH₄ was used. This will generate sodium (and potassium) adducts of the detected species, additionally to the protonated species.

In the absence of an internal standard or (stable) heavy isotope labeling standard, all the combination of proton/sodium/potassium adducts have to be monitored for each charge state of the muropeptide.

Moreover, in-source degradation of the muropeptides such as anhydro-muropeptide (neutral loss of water) and GlcNAc losses will be formed in source and have to be considered too.

We have introduced raw m/z in SI (Fig. S2a, S3b).

We agree with the reviewer that we are operating with a complex sample and selected procedures may lead to formation of adducts. In analysis presented on Fig. 3c and Fig. S3, we followed only dominant peaks of the spectrum of muropeptides generated by mutanolysin, which are well documented in the literature (i.a. Kühner et al., 2015, Monteiro et al., 2019), namely monomer (1253Da), dimer (2417 Da) and trimer (3579 Da). We did not find any dominant peaks of adducts of neither Na (+23 Da) nor K (+39 Da) in analyzed spectra, so we concluded they comprise a minor fraction in terms of relative abundance. For clarity of conclusions and not to draw false conclusions stemming from wrong identification of the adducts, we did not follow their digestion. To stay transparent with the readers, we included this information in Materials and Methods (line 527).

Figure 2e: what does the dashed lines stand for ?

It seems to be related to the basal level of muropeptides in the absence of the enzymes ?

We are sorry for this confusion, we changed the way of presenting the data to be more clear.

L151:

This is meaningless without the description of the monitored m/z ions as specified in my previous comments for line 119

m/z spectra were added.

L159-160: Please rephrase. "Complex peak pattern" does not properly describe a 2D NMR data

We changed the sentence to "Mature Lss digests sacculi and produces a multipeak spectrum, similar to muropeptide digestion products."

L161: The authors are still referring to Lss ?

Yes, for clarity we changed the sentence to: "However, no product of the reaction could be detected using only the Lss catalytic domain. "

The reading of the 1st paragraph under the section "Substrate complexity impacts the activities of LssCD and LytMCD" are sometimes confusing and deserve some rephrasing for clarity

This has been rephrased and corrected with tracked changes.

L161: "However, no product of the reaction could be detected using only the catalytic domain" I think the Fig. 2d referenced to this sentence and not the previous one in the text
We are sorry for this confusion, this data was not shown, but has been introduced in the text now. Additionally, we added information in the text indicating which LssCD reaction was presented on Fig. 3a (line 166).

L162: I think the end of the sentence is referring to the Fig. 3A that should be mentioned in the text "This is in clear contrast to muropeptide processing, completed by both mature Lss and its catalytic domain alone"

This has now been included. In addition, there was a mistake in the sentence, we meant "PG" not muropeptides, therefore the sentence has been corrected.

L163: this sentence has no direct link with the previous one. Please rephrase it, e.g. by adding "In an additional experiment, we increased the LssCD concentration (...)"

This has been rephrased.

L164: Please rephrase. "Multi peak detection" is meaningless because it is the purpose of the NMR: see the coupling of the nuclear spin due to the molecular orbital shielding...

You can rephrase as follow: "we observed in the NMR spectrum the peaks pattern corresponding to the digested pentaglycine bridge"

This has been rephrased.

L166: change "demonstrated" by "this is also supporting".

This has been changed.

L178: 10% of what ? Of the NMR signal ? Something else?

I propose that the authors rephrase the sentence as follow:

"S. aureus PG displays a high degree of cross-linking, as uncross-linked muropeptides (monomers) account for less than 10%" to "More than 90% of the muropeptides are involved to cross-linking of PG"

Thank you for this suggestion, this has been modified accordingly.

L184: sentence revision: there was a “the” missing

“(…) enzymes digest mucopeptide monomers, but only LssCD digests the mucopeptide dimers and trimers”

This has been added.

L187 – 194: Here is the real problem with this work concerning the LC-MS data:

In the total absence of Material & Method section concerning the HPLC and MS experiments, and in regards of the cited literature, one has to assume the HPLC method used by the authors was a reverse phase HPLC method, inspired from the paper of Kuhner and coworkers.

There is no description about the adaptation required to make it working in the context of coupling MS with HPLC.

If my initial guess is correct, in this case, there is no way that the monomer, dimer, trimer or tetramer of soluble mucopeptide fragments to be eluted within less than 12 minutes.

This is clearly shown in the paper from Kuhner and coworkers in 2014 (DOI 10.1038/srep07494) and confirmed by Boulanger and coworkers in 2019 (DOI 10.1002/elps.20190047) for closely related mucopeptides that the ones investigated by the authors.

In other word, all multimeric species detected below 12min using the same HPLC method from the work from Kuhner and coworkers correspond to chromatographic fractions that experienced carry-over between LC runs! This is obviously something we do not want, never!

In this regards, the authors have to fully describe their method and justify how/why the multimeric mucopeptides can be eluted before 12 minutes, or revisit their data and the text before publication.

In my opinion, the only reliable LC-MS data is the absence of detection of some mucopeptides (figure 3C) but the detection limit have to be assessed, or at least properly estimated or discussed.

By chance the mucopeptides were detected for the other experimental conditions, meaning that the limit of detection is not critical figure of interest, and that the relative comparison of the normalized LC-MS signals still support well the discussion of the authors.

We agree with the reviewer that the methodology of UPLC/MS analysis was not adequately presented, making it difficult to follow how the procedure was performed. The detailed protocol has now been described in Materials and Methods.

To analyze samples, we used a mass spectrometry service provided by MS Lab (Mass Spectrometry Laboratory, Institute of Biochemistry and Biophysics, Polish Academy of Sciences). The service run our samples with standard procedure, which they established for separation of proteinous samples using acetonitrile gradient by applying phase A (0.1 % formic acid in H₂O) and phase B (0.1 % formic acid in acetonitrile) during 12 minutes run (100 % A for 2 minutes, 0 %- 40 % B for 6 minutes, 40 %-90 % B for 2 min and 99 % A for 2 minutes). We agree with the reviewer that this does not give a well resolved chromatogram of mucopeptides, therefore in the analysis we processed the entire spectrum and searched for selected molecular masses. Following that, to emphasize differences between the samples, we follow intensity changes in m/z and mass spectra. Due to the high abundance

of the detected muropeptide variants, we changed presentation of the data from relative to absolute intensity values.

Without replicates and estimation of the standard deviations, and in the absence of reliable and robust quantitation strategy, MS data from the figure 2 has to be interpreted with extreme caution.

We recalculated these data, and included them as m/z and processed mass spectra (Fig. S3), showing that 1182 Da peak is a dominant mass in LssCD processed sample while in mutanolysin processed sample (control) it was 12 times less abundant. Samples were prepared from the same stock and loaded in the same amount, therefore we treat this result as a meaningful increase of the MS readout matching in mass of possible reaction product. 1125 Da mass was also one of the dominant peaks in LssCD spectrum and it was 10 times more abundant than in the control sample. Two empty runs were performed to avoid column carryover.

To avoid inaccuracies stemming from presenting the exact values, we transformed our data into a heatmap, which presents the ranges and underlies qualitative differences between the samples (Fig. 2e, Fig. 3c).

L198-200: In regards to the Figure S4, my opinion is that the mature Lss enzyme had the same clearance efficiencies (in terms of turbidity reduction which correspond to the lytic activity) for the strongly cross-linked peptidoglycan (PG) and the PG with lower reticulation

*We agree with the reviewer, this paragraph has been corrected to: "We performed an analogous assay using the mature version of this enzyme, which strongly binds CW through its binding domain, and found that **mature Lss displays similar clearance efficiencies for strongly cross-linked PG and the PG with lower reticulation.**"*

L200: Consequently you can delete "In contrast" because LytMCD and mature Lss acted similarly whatever the level of reticulation of the PG, always according to the data of fig. S4 *This has been modified accordingly.*

Figure 3

legend of Fig3b: I think it will be more clear and relevant to use "mutanolysine alone" instead of "no enzyme" *This has been corrected.*

figure caption 3b: there is a typo, you wrote "resissues" instead of "residues" *This has been corrected.*

y-axis of figure 3c: you should use "Residual relative amount of muropeptides identified by HPLC-MS" instead of "relative intensity of the mass signal". Also adapt the discussion in the plain text accordingly. *This information was included in the text (lines 178-179). We modified the presentation of this figure to show data as intensities extracted from the MS spectra.*

caption of 3c: did you mean "Total Ion Current (TIC) within the chromatographic peak" instead of "to the sum of all the intensities" ? *Again, due to modification of the data presentation, we have deleted this sentence.*

Fig3c: "The readout of muropeptide dimers identified in the LytMCD processed samples is higher as compared to unprocessed sample, which could stem from processing of non-analysed muropeptide multimers"

In the absence of replicates, it is more reasonable to assume that it is related to the uncertainties of the experiment, i.e. your standard deviation, especially in the absence of any kind of (internal) standard

It will reasonably be better to show normalized data in regard of the signal of “Muropeptides” for each case (i.e. monomer, dimer, trimer) than a sum of signal which are meaningless in regard of the MS method, assumed to not use any (semi)quantitation with (internal) standards

Note that the remarks is also true for the MS data in figure 2.

Following the reviewer's advice, we reformulated the presentation of the analysis as explained before (answer to remark for lines 187 – 194). In the figure caption, we have included a sentence explaining increased intensity for LssCD and LytMCD samples as compared to mutanolysin treated sample according to the reviewers suggestion: “Increased intensities for the masses of monomer detected in LytMCD and LssCD samples are related to uncertainties of the experiment.”.

L231:

please use “by” instead of “in” in the sentence “ (...) distinct digestion patterns of the multimeric muropeptides obtained BY MS analysis challenge the assumption that (...)”

This has been corrected.

L235: I think you meant Fig 2a and Fig 4a (which refer to the structure of the “substrate analogue” containing one serine instead of a glycine in the cross-linker peptide of (Gly)₅

That's correct, this was added.

L258: Please use “suggesting” instead of “implying”. I think the data are not strong enough to affirm such statements about the loop, at least at this stage of the “Results” sectioning

This has been corrected.

L265: What “Ext. Data Fig 1 and 2” stand for ? Did you mean

424312_0_data_set_413380_rsqp07.txt and 424312_0_data_set_413381_rsqp84.txt ?

We are sorry for this confusion, we have submitted extended figure files to the online system with these names: “Ext_data_1_LssCD_substrate_1st_best_structure” and “Ext_data_2_LytMCD_substrate_1st_best_structure”. Probably this was changed to new names automatically.

L267: I think you meant Fig. S9 instead of Fig. 9

Sorry for this mistake, it was corrected.

L323: I am pretty sure that koff is a kinetics constant, consequently, the unit of koff are in min⁻¹ and not in concentration (μM in the manuscript),

kd would be in M⁻¹ and kon would be in M⁻¹.min⁻¹

By the way, even your sentence strongly support a scale time unit instead of concentration.

Please correct or rephrase

This has been corrected.

L348: What “Ext. Data Fig 1 and 2” stand for ? 424312_0_data_set_413382_rsqp9j.txt and 424312_0_data_set_413383_rsqp9j.txt ?

Again, the files have been submitted as
“Ext_data_3_LssCD_hexadimuropeptide_1st_best_structure” and
“Ext_data_4_LytMCD_hexadimuropeptide_1st_best_structure”. We’d like to ask editor to
check this.

Additionally, I get some trouble with the visioning of the PDB structure where some of the
3D part of the protein were missing (PyMol v2.3.0 under Linux).

Honestly, I did not had the time to investigate the structure of these files and determine if it
was a problem related to my version of PyMol or with the input files.

*We have inspected the pdb file using two different PyMol versions (PyMol v2.3.3 under
Windows and PyMOL v2.4.2 under Macintosh) and the protein is visible.*

L451: Material and Methods

As mentioned above, there is absolutely not detail about the liquid chromatography and the
mass spectrometry detection, neither in the supporting information.

The type of chromatographic pumps and column

Waters Acquity UPLC System

and the stationary phase, dimension, mobile phases,

Acquity UPLC Protein BEH C4, 300A, 1.7um, 1mmX50mm (cat no. 186005589)

gradient, flow rate,

*Acetonitrile gradient performed by applying phase A (0.1 % formic acid in H₂O) and phase B
(0.1 % formic acid in acetonitrile) during 12 minutes run (100 % A for 2 minutes, 0 %- 40 %
B for 6 minutes, 40 %-90 % B for 2 min and 99 % A for 2 minutes). The flow rate was 100 μ L
min⁻¹.*

injection volume and condition, *10 μ L*

if any guard column or trap column was used ? *It was not used.*

The mass spectrometer is also not described. There is a mention of a Time of Flight in the
legends of some figures in S.I. Please describe properly the instrumentation, as well as the
ionization mode (assumed to be electrospray positive ionization mode),

Synapt G2 (Waters) operating in ESI positive ionization mode

source temperature and desolvation gas flow rate,

*Source temperature was set to 80° C and desolvation temperature to 150 °C. The flow rate
of the desolvation gas was 650 l/h.*

the mass range,

100-3000 m/z

what was the average mass accuracy and/or the RMS of the mass accuracy within the mass
range

We have calculated the average mass accuracy using webtool

*(https://warwick.ac.uk/fac/sci/chemistry/research/barrow/barrowgroup/calculators/mass_error_s/) and based on observed/theoretical mass for monomer, dimer and trimer, we estimate that
the spectrometer provided by the mass spectrometry facility gives the error of around 100
ppm and 0.3 Da.*

What were the monitored m/z ?

m/z graphs were provided in Fig. S2a and Fig. S3b

Is any standard was used to perform the (semi)quantitation ? Is the in-source fragmentation products were monitored together with the precursors ions ? *This was not performed.*

What was the versioning of MassLynx ?

MassLynx V4.1

Did you smooth the peak? What was the integration algorithm and the settings ?

We don't smooth peaks. Mass spectra were deconvoluted using the MaxEnt3 algorithm provided with the Masslynx software, with parameters set for minimal molecular mass of 200 Da, maximal molecular mass of 8000 Da and maximal number of charges 5.

Is any blank injected between runs ?

Yes, two blanks in between.

I have to assume the author performed LC-MS coupling on-line, which can be quite challenging without some good expertise for the LC-MS investigation of mucopeptide. Nothing in the manuscript demonstrated or support such mastering and the reader need to have "blind and absolute" confidence.

Without these basic information, there is no possible evaluation of the LC-MS data

This procedure has been now described in details in MM section.

In the Material and method, there is some abbreviation that are never used. Please correct the text accordingly.

To the contrary, some of them are used but never described, e.g. WTA which I assumed to mean "Wall Techoic Acids"

This has been corrected.

"Cloning, production and purification of proteins"

L 496: What kind of Amicon did you used ? 1KD, 3KD, 10KD, 10KD ?

We used 3 kDa cut-off, this was added.

"PG purification"

The author barely justify the use of the strain SH1000. Please refer to your text or mention briefly the type of produced mucopeptides that you used in your study as a control.

*This was explained: "¹³C-¹⁵N-labelled PG of *S. aureus* strain SH1000 used for NMR digestion experiments was prepared as described previously⁴⁷."*

Authors have to motivate why the first 12 minutes of the LC runs are showed while there is almost no chance that all the mucopeptides could be eluted from a reverse phase column even without the use of a pairing agent such as trifluoroacetic acid (or why not heptafluorobutyric acid).

The answer for this question was provided in the comment L187 – 194.

L500: "The lytic assay on Remazol-dyed PG was performed overnight with 1 μ M hydrolase at 37 °C in 50 mM glycine, 100 mM NaCl pH 8.0 buffer against stained PG of OD₆₀₀ ~ 6"

Please rewrite, this section is unclear. Moreover I think you meant OD~ 0,6 instead of 6

This has been rephrased to: "1 μ M hydrolase and remazol-dyed PG were mixed together and the reaction topped up to 400 μ l with 50 mM glycine, 100 mM NaCl pH 8.0 buffer. Final OD₆₀₀ of PG was ~ 0,6. PG digestion was performed overnight at 37 °C. ". Indeed, we meant 0,6. We have also added a missing reference for this method.

L515:

Did you mean enzyme instead of protein ?

Yes, this was corrected.

L546:

Change “analysed” for “processed”

This was changed.

L549:

What is the “in this area” you mentioned ? Please rephrase and complete this sentence.

This has been completed: “To quantify the digestion products, the intensity of each peak was quantified and normalised to the sum of the intensities of all peaks in the 114-117 ppm ¹⁵N-chemical shift region.”.

Reviewer #2 (Remarks to the Author):

In this manuscript, the authors conducted a comprehensive analysis of the enzymatic activity and specificity of two closely related M23 hydrolases, lysostaphin (Lss) and LytM, utilizing solution NMR techniques, Chemical Shift Perturbations (CSPs) of amino acid resonances, and magic-angle-spinning solid-state NMR (ssNMR). The findings revealed distinct binding patterns of the catalytic domains of these enzymes to the cell wall and differential effects of peptidoglycan cross-linking on their activity.

The manuscript offers a compelling strength as the authors successfully constructed a detailed model that maps the interaction between the M23 hydrolases and cell wall substrates at the amino acid level. Also, the manuscript is clearly written, and the provided data generally substantiate the proposed model. Nonetheless, I have identified a few concerns regarding this study, which are outlined below.

1. The physiological relevance of the findings in this study raises some questions. While the study primarily focused on the catalytic domains, Fig. 3 highlights the crucial role of the cell-wall binding domain (CBD) in enzyme activity. Consequently, it remains unclear to what extent these results can be extrapolated to intact cell wall hydrolases.

We agree with the reviewer that presence of CBD domain affects the lytic performance of the Lss enzyme. However our results show that Lss and LssCD give similar digestion patterns, although the intensity of the obtained products is different (Fig. 2d and Fig. 3a). Based on that, we believe that CBD affects the enzyme kinetics but does not change its specificity.

Therefore, we think that our insights into enzyme orientation and interaction with PG could be still translated to physiological conditions even though we are using a model system of catalytic domains alone. Our main motivation for this study was to gain insights into the M23 peptidases activity and cell wall binding. CBD of lysostaphin is thoroughly covered in literature, also by structural studies showing its interactions with PG fragments (e.g. Tossavainen et al., 2018, Gonzalez-Delgado et al. 2021), therefore we found it less prioritizing to investigate its interaction with the cell surface. To keep our aims transparent for the reader, we included this rationale at the beginning of the manuscript text: “We used a simplified model using catalytic domains alone to gain insights into interaction of two evolutionary conserved enzymes with bacterial cell wall.” (lines 73-74).

2. The binding model does not explain why the activities of Lss and LytM are affected by cell-wall cross-linking.

Binding model indicates that LytM forms more molecular contacts with complex PG than Lss, i.e. LytM loop 1 is positioned in proximity to neighboring cross-bridge and forms H-bond with uncrosslinked stem-peptides. While LytM is attached well to the cell wall via CBD_{Lss} domain (AuresinePlus), it hydrolyses S. aureus displaying reduced PG cross-linking faster than S. aureus with regular cross-linking. Combining these two observations, we think that cross-linking serves as the control mechanism to limit activity of LytM to cell wall expansion. In this scenario, LytM would act in the regions of the cells which display reduced crosslinking due to activity of other S. aureus autolysins or mechanical extension of PG strands.

In contrast, Lss is a highly effective enzyme (bacteriocin), which evolved as a potent antimicrobial warhead. Lss forms less molecular contacts with PG, which in the proposed scenario alleviates substrate constraints and thus contributes to increased processivity of the enzyme. Lytic assays on the reduced/regular cross-linking S. aureus variants indicate that the enzyme acts faster on the regions with regular cross-linking, although in its mature, two-domain form, this difference became very minor. Therefore, we concluded that the limited number of molecular contacts formed between Lss and PG makes this enzyme to act in location and cycle-independent manner.

Since a similar comment was raised by reviewer 3, we completed the manuscript by providing an extended paragraph linking binding models with the role of cross-linking and biological significance of this regulation in the discussion (lines 421-438).

3. There appears to be a discrepancy between Fig. 3d and Fig. S4 (LssCD vs Lss) in the manuscript. While Fig. S4 indicates that Lss equally digested cells grown in both MHB and MHB+ sub-MIC oxacillin, Fig. 3d suggests that Lss more efficiently digests cells grown in MHB compared to those grown in MHB+ sub-MIC oxacillin.

Sorry for this confusion. We included in the figure caption the missing information on the reaction time point presented on Fig. 3d (30 minutes).

Activity of mature Lss for cells grown in MHB and MHB+ sub-MIC oxacillin are very similar, which is indeed more evident by looking at time-course (Fig. S4). This point was also addressed by reviewer 1, so we corrected the paragraph to: "We performed an analogous assay using the mature version of this enzyme, which strongly binds CW through its binding domain, and found that mature Lss displays similar clearance efficiencies for strongly cross-linked PG and the PG with lower reticulation." (lines 195-198).

4. The turbidity reduction assay depicted in Fig. 3b and d would benefit from additional clarification. It is puzzling that LytMCD failed to hydrolyze sacculi in Fig. 3b, yet it appears capable of lysing S. aureus in Fig. 3d. Considering that S. aureus is known to produce numerous cell wall hydrolases (at least 18), it is plausible that other hydrolases present on or within S. aureus cells might contribute to the observed cell lysis. To address this concern, it may be advisable for the authors to include a no-enzyme control in their experimental design, which would help elucidate the specific contribution of LytMCD in the context of cell lysis.

Following reviewers advice, we included the no-enzyme controls in our experiment design. We monitored lysis of cells due to activity of other autolysins in two buffer conditions used for the turbidity reduction assay, namely with and without 100 mM NaCl. We observe minor cell

lysis in all experimental variants: 1% of S. aureus with regular crosslinking lyse due to autolysin, whereas no lysis was observed for S. aureus with decreased crosslinking in the buffer without addition of salt. These results were added to Fig. S4b. To clarify this matter, we included this information in the manuscript text: “Cell lysis due to activity of autolysins was monitored and comprised a minor fraction of monitored lysis (up to 1% of all cells), indicative for that the observed cell lysis derives from the activity of externally added enzymes (Fig. S4b).” (lines 203-205). In addition, we include in legend caption the information that the results presented for lytic activity assays were expressed in relative to no-enzyme controls (line 224).

Reviewer #3 (Remarks to the Author):

The manuscript by Razew et al. describes an investigation of the binding and substrate specificity of two members of the M23 hydrolase family that cleave the penta-glycine crossbridge peptide in *S. aureus* peptidoglycan (PG). These include the *S. aureus* LytM protein that helps to remodel PG during growth and division, and the Lss enzyme that is secreted by *S. simulans* to eliminate bacterial competitors. Prior structural studies have deduced the mechanism of glycine peptide binding by this enzyme class. Here, the authors seek to gain insight into how these enzymes engage their substrates within the natural context of the bacterial cell wall. This is very challenging as PG is a high molecular weight and heterogeneous structure. Using mutanolysin degraded fragments of PG as enzyme substrates the authors convincingly localize the primary site of cleavage (between the 4th and 5th glycine residues). The authors also present data that reveals that the enzymes digest multimeric forms of PG differently, and that they bind to PG differently. Finally, HADDOCK derived models using NMR chemical mapping data suggest that they engage their substrates differently which may explain their distinct substrate preferences.

Strengths of this work include the application of a broad set of biophysical tools to investigate how biologically important PG hydrolases recognize their substrates in the natural context of the cell wall. The data seem to nicely support the idea that distinct structural features in the enzymes may alter the contacts each enzyme makes with PG components that surround the cross bridge peptide. However, there are major weaknesses.

First, it is unclear if the molecular models presented in this paper are accurate to the level of detail that is claimed. In particular, figures 4D and 5F show specific enzyme-PG interactions that seem highly speculative given the low-resolution nature of chemical shift mapping data and the fact that the 3D structure of the PG can only be assumed. Thus, I am not convinced that they have determined at high resolution the details of how these enzymes engage PG sacculus (a central claim made in the abstract).

We agree with the reviewer that the final information provided by CSPs monitoring through liquid state NMR (although monitored at the residue level) provides moderate resolution information depicted by data-driven modeling. To minimize speculative aspects of modeling process we (1) generated optimal energy substrate and hexadimuropeptide models, (2) analyzed at least four structures of the best energy cluster provided by HADDOCK (Fig. S7 and S12), and (3) calculate probability of contact formation for all best energy structures using Liplot+ (Table S5, Fig. S8-9, S14-15). Following reviewers suggestion and to avoid providing misleading information to readers, we modified the abstract to: “Here, we provide structural insights into the interaction between short peptidoglycan fragments and the entire sacculus with two evolutionarily related peptidases of the M23 family (...)” (lines 14-15). In addition we

modified text throughout the manuscript to underlie the low resolution nature of CSPs mapping and keep transparent with the reader on the limitations of generated models (lines: 78, 303-304, 353, 373-374, 401-402, 451).

Second, it is unclear if the mutanolysin degraded PG is an accurate surrogate for the intact cell wall that surrounds the microbe. Thus, making biologically relevant conclusions about how the enzymes function to degrade the cell wall seems speculative.

*We are sorry for this confusion - apart from mutanolysin digested PG, which was used for digestion experiments and NMR titrations (Fig. 2 and Fig. 4), we also used intact cell walls (purified PG containing cell wall teichoic acids) as the substrate for the digestion reaction (Fig. 3a, b) and as insoluble material to pull down the enzymes (Fig. 5). In the latter case, we tracked the interaction of the enzymes **with intact PG** by comparing NMR spectra of free and bound enzymes. This allowed us to identify residues engaged in the interaction. The information on which type of PG material (mutanolysin digested vs. intact PG+teichoic acids) has been now underlined throughout the manuscript.*

Third, although the results are described in detail it is unclear why they are significant. Why is it important that the Lss and LytM enzymes engage and digest PG differently? Are the results predictive of enzyme function in other microbes?

*That is an interesting point. The observation that Lss and LytM engage and digest PG differently is consistent with distinct physiological roles that these enzymes play in bacterial metabolism. LytM is produced by *S. aureus* to digest its cell wall in a timely and spatially controlled manner (autolysin). Therefore it requires tight control, which takes place on several levels (transcriptional, post-transcriptional, posttranslational). It is also known that this enzyme is produced in high abundance, therefore requires mechanisms of tight posttranslational control of its activity. One of the means for that is the presence of an N-terminal occluding region, which occupies substrate binding groove and blocks enzyme's active center. In contrast, Lss is produced by *Staphylococcus simulans* to eliminate competing staphylococcal species and to clear its ecological niche. Therefore, Lss is a highly effective enzyme (bacteriocin), which evolved as a potent antimicrobial warhead. Here, we made a new observation which fits the present state of knowledge. Taking into account that LytM is under tight tuning, it is not surprising that it forms more molecular contacts with the cell wall. This possibly serves as another means to limit its activity to a specific physiological state of the cell, namely cell wall expansion. In contrast, Lss forms less molecular contacts with PG, which in the proposed scenario alleviates substrate constraints and thus contributes to increased processivity of the enzyme. Taking into account that PG hydrolases acting as either autolysins and bacteriocins are commonly found across bacteria, it is possible that the same phenomena could be observed in other microbes.*

Since a similar comment was raised by reviewer 2, we completed the manuscript by providing an extended paragraph explaining the biological significance of our findings in the discussion (lines 421-440; 454).

REVIEWER COMMENTS

Reviewer #1 (Remarks to the Author):

Please see my report in attached file (docx or PDF files).

Reviewer #3 (Remarks to the Author):

The manuscript is acceptable for publication

Razew and coworkers, *S. aureus* sacculus mediates activities of M23 hydrolases

Nature comm., revision 2

Main remarks:

First I have to thank the authors to take in account most of the suggestions of the reviewers.

I appreciated the efforts of the authors to properly describe the MS experiments. This is incidentally confirmed my feelings about how not too far we can exploit these MS data. The authors tried to use a quantitative aspects of the MS data to support some of their hypotheses, and I am still claiming that presence/absence of some targets is the best the authors can claim without proper normalization of the MS data. Additionally the mass accuracy produced by the authors would force the exploitation of the data even more carefully, especially with poorly chromatographically resolved peaks. Nevertheless there is still some efforts to provide before publishing the submitted article.

During the second reviewing session of the revised article, I meet some of the remarks of one of the other referee claiming about some speculative arguments concerning the structure-activity relationship of Lss and LytM.

I also have now the feeling of some « conflicting » results that I am not able to interpret properly to explain some apparent differences of substrate specificity. The nature of the enzymatic products and the specificity of the cleavage sites cannot explain alone the difference between Lss and LytM in regards of the degree of reticulation of the peptidoglycan (PG).

At this stage, I still have some concerns about the mass spectrometry data.

As I requested in my first report, the authors provided some information about the M&M of the MS analyses.

Nonetheless, my concerns about the quantitative aspects (substrates and products) in the paper are not yet fixed. Indeed, in the absence of normalization of the injected amount of biological material, i.e. mucopeptides and the enzymatically released products in the different condition, could be hardly compared.

The fluxes of substrate(s) - - -> products is not yet known.

As an attempt to looking for about a potential normalization based only on the data provided by the authors in the main paper and the supporting information, I tried adding all the MS intensities of the monitored species which do not guaranty any proper normalization of the data.

Using this strategy, I can only conclude that LssCD would significantly (1 ordre of magnitude) produce more G₅-stem-G₁₋₄ and G₅-stem-G₁₋₃ than the mucopeptide condition (table S1/S2).

Please note that in the absence of the monitoring of the water soluble products G₂-G₃-G₄ and G₂-G₃ or G₃-G₄ oligopeptides (or the “acceptor” stem), this was quite an unsafe attempt of normalization strategy, but a “better than nothing” solution.

My opinion is that the absolute intensities of the detected ions should be normalized e.g. in terms of microgram of injected dry mass of pellet (assumed to be purified peptidoglycan), before trying to interpret the efficiency of the LssCD or LytMCD.

That is why I am still claiming the authors can interpret the data in terms of detected/not detected species, but I will not recommend to interpret further the MS data, instead of claiming that LssCD or LytMCD is more or less efficient in regards of the substrates.

Once again, the work is (in my opinion) mainly supported by NMR data and the HADDOCK modeling. Nonetheless I am not still convinced that the data presented by the authors unambiguously explain the molecular mechanisms of Lss(CD) and LytM(CD), or explain the substrate specificity of this enzymes.

I would also suggest to add a perspective paragraph claiming that they should validate or cross-check their hypotheses by additional set of experiment such as cross-linking MS, hydrogen/deuterium exchange (HDX) by MS, or HDX NMR.

Indeed, to explain the structure-activity relationship, I would be not surprised if the binding domain (after binding the sacculus) would affect the 3D conformation of the catalytic domain of LssCD or LytMCD, explaining the specificity of these M23 hydrolase depending the reticulation level of PG.

Main article:

Abstract

suppress line 20-21, to fuse with line 18

... we propose a new model in which peptidoglycan cross-linking affects the activity, selectivity and specificity these two structurally related enzymes differently.

L62-63: (...), is the native species of Lss.

L87 Figure caption Fig 1: Please change this sentence: “schematic representation of *S. aureus* sacculus and LssCD (orange) and LytMCD (yellow) digestion site”

L92: authors claimed that the cross-bridges are indicated with an arrow, but this was not the case. Please update the caption or the figure accordingly.

Figure 2: During the second reviewing, I realize that the box area from the 2D NMR data in the ^{15}N 118-128ppm and ^1H 7.6-8.2ppm differ significantly. Is the authors have any idea of the possible annotation of these peaks? Could be some evidences of cross digestion products by (e.g. other) M23 hydrolases in the spectra? Maybe these data contains additional evidences helping the interpretation of the structure-activity relationship.

What does the dark grey background mean in Fig S2c “muropeptides”?

Fig 2e: Even if we discard my previous comments in the beginning of my report: It would be better if the authors compared at least the absolute MS signals before and after the action of the LssCD (or LyMCD “enzymes), the heatmap would make more sense. Without any normalization, these data are meaningless.

In the present work, the only normalization relies on the sample preparation of the authors, which performed the bacterial growth until 0.6 O.D. Nonetheless, nothing ensures that the amount of produced full-length PG products or degradation products are produced at the same level between experiments, unless the authors produced only one big batch of PG extract. In this case, it should be claimed by the authors in the M&M.

L172-177: Authors claimed that LytMCD does not release soluble products. Nonetheless LytMCD releases oligoglycines that were not monitored by NMR or MS, which are soluble released products.

L180: Proposition of rephrasing. “We then hypothesized that LytMCD could cleave the cross-bridge without releasing soluble fragments.”

L200: Please remind to the reader that some of your mass spectra are deconvoluted spectra while other are raw (zoomed) mass spectra. I am a MS expert, si it is obvious for my eyes but it is not for most of your target readers.

L202: Due to the use of an excess of NaBH_4 for the reduction of your PG fragment (because of the CMS analysis later), the amount of Na (and K) adducts will be still very significant, even if the

samples were injected in an LC column before MS detection. Consequently this will strongly affect the result of (semi) quantitation of the different targets by MS

L204-206: I think instead that if the cross linking of the PG decrease, then the amount of especially monomers, as well as dimers, and trimers would increase.

To the contrary if the cross-linking increase, then the amount of monomers, dimers, and trimers will decrease. I suggest to revisit this section.

L256: I am still requesting to the authors to explain/justify the use of *S. aureus* SH1000 strain. What is the specificity/properties of the PG produced by this strain? Please provide some context somewhere in the introduction, or M&M or the plain text.

L305/ Please mention in the figure caption that the residues are numbered following the nomenclature provided in Figure S7.

L412: Please correct that the NMR spectra are ^1H ^{13}C ^{15}N labelled LssCD instead of ^2H ^{13}C ^{15}N labeled LssCD.

L438: Please mention that you (probably) refer to the Fig. 3C (multimeric forms of mucopeptides are digested efficiently by LssCD but not by LytMCD).

L548: I assumed that you used a Synapt G2 HDMS. If not please specify which series of SynaptG2 you used.

L553: 650 L/h, not l/h

Supporting information :

Disclaimer:

During the reviewing process, the reviewer does not have any access to MS software tools. All was manually processed and mistakes cannot be excluded.

Figure S1 : For clarity, I would better suggest to edit the figure caption as « Chemical structure of the non cross-linked peptidoglycans and oligoglycin peptides (...) »

Note that neither chemical formula or masses or provided, in contrast with what is claimed in this caption. If the authors want to present MS data, chemical formula is a more usefull information than « masses ». Indeed we do not know what kind of masses are provided : neutral compounds, ions, adducts, and charge state ... ?

Figure S2 : The exact mass I computed differ from the mass provided by the authors. Moreover the m/z provided by the authors change for the same species. Please provide uniform information.

Consequently one can deduce that the average mass accuracy of the MS data is about 50ppm (RMS), which is surprisingly high for a Synapt G2 mass spectrometer, even for external mass calibration. This is suggesting that the intensities of the NaI calibrant do not fit the intensity of the ions detected during UPLC MS experiments.

If I reinterpret the MS mass spectrum provided in SI, I can deduce species having 1 Da mass lower that the mass suggested by the authors (mono-isotopic mass would be 626,39 instead of 627,7x presented by the authors, $z=+2$).

Additionally, I have to assume that the MS instrument is actually a Synapt G2 HDMS instrument.

So please, explain how the authors determined the mass $[M+H]^{x+}$ presented in the SI.
In my opinion, authors should provide the theoretical exact mass of the monitored species in regards of the showed mass spectra.

Figure S3: Globally same remarks

Figure S4: Provide the signification of MHB in the SI

Figure S14 and S15: The residues and their respective numerotation are barely readable. This is making hard for the reader to conceive how would/could be the interaction between the residues of the enzyme and the substrate, especially in mentally reconstructed 3D structures.

Table S1 and S2 have a typo in « muropeptides »

We thank reviewer for cautious re-reading of the manuscript. We carefully included all the comments to provide this improved version of the manuscript.

Razew and coworkers, *S. aureus* sacculus mediates activities of M23 hydrolases

Nature comm., revision 2

Main remarks:

First I have to thank the authors to take in account most of the suggestions of the reviewers. I appreciated the efforts of the authors to properly describe the MS experiments. This is incidentally confirmed my feelings about how not too far we can exploit these MS data. The authors tried to use a quantitative aspects of the MS data to support some of their hypotheses, and I am still claiming that presence/absence of some targets is the best the authors can claim without proper normalization of the MS data. Additionally the mass accuracy produced by the authors would force the exploitation of the data even more carefully, especially with poorly chromatographically resolved peaks.

Nevertheless there is still some efforts to provide before publishing the submitted article. During the second reviewing session of the revised article, I meet some of the remarks of one of the other referee claiming about some speculative arguments concerning the structure-activity relationship of Lss and LytM.

I also have now the feeling of some « conflicting » results that I am not able to interpret properly to explain some apparent differences of substrate specificity. The nature of the enzymatic products and the specificity of the cleavage sites cannot explain alone the difference between Lss and LytM in regards of the degree of reticulation of the peptidoglycan (PG).

At this stage, I still have some concerns about the mass spectrometry data.

As I requested in my first report, the authors provided some information about the M&M of the MS analyses.

Nonetheless, my concerns about the quantitative aspects (substrates and products) in the paper are not yet fixed. Indeed, in the absence of normalization of the injected amount of biological material, i.e. mucopeptides and the enzymatically released products in the different condition, could be hardly compared.

The fluxes of substrate(s) - - - -> products is not yet known.

As an attempt to looking for about a potential normalization based only on the data provided by the authors in the main paper and the supporting information, I tried adding all the MS intensities of the monitored species which do not guaranty any proper normalization of the data.

Using this strategy, I can only conclude that LssCD would significantly (1 ordre of magnitude) produce more G₅-stem-G₁₋₄ and G₅-stem-G₁₋₃ than the mucopeptide condition (table S1/S2).

Please note that in the absence of the monitoring of the water soluble products G₂-G₃-G₄ and G₂-G₃ or G₃-G₄ oligopeptides (or the “acceptor” stem), this was quite an unsafe attempt of normalization strategy, but a “better than nothing” solution.

My opinion is that the absolute intensities of the detected ions should be normalized e.g. in terms of microgram of injected dry mass of pellet (assumed to be purified peptidoglycan), before trying to interpret the efficiency of the LssCD or LytMCD.

That is why I am still claiming the authors can interpret the data in terms of detected/not detected species, but I will not recommend to interpret further the MS data, instead of claiming that LssCD or LytMCD is more or less efficient in regards of the substrates.

We rephrased the text and changed presentation of the figures 2e and 3c to interpret the data in terms of detected/undetected (products analysis: lines 124-127; substrate analysis: 181). We recalculated these data by expressing in relative to all MS intensities of the monitored species as suggested by reviewer 1.

Once again, the work is (in my opinion) mainly supported by NMR data and the HADDOCK modeling. Nonetheless I am not still convinced that the data presented by the authors unambiguously explain the molecular mechanisms of Lss(CD) and LytM(CD), or explain the substrate specificity of these enzymes.

I would also suggest to add a perspective paragraph claiming that they should validate or cross-check their hypotheses by additional set of experiment such as cross-linking MS, hydrogen/deuterium exchange (HDX) by MS, or HDX NMR.

Indeed, to explain the structure-activity relationship, I would be not surprised if the binding domain (after binding the sacculus) would affect the 3D conformation of the catalytic domain of LssCD or LytMCD, explaining the specificity of these M23 hydrolase depending the reticulation level of PG.

Based on our own experience, and that of our collaborators, on peptidoglycan, studies using other techniques are often difficult and complex to analyse. In this type of dynamic system involving a protein and a polymer, proton exchange or cross-linking methods perform poorly, partly due to the low affinity of the complex. This seems to us to be one of the interests of the manuscript, which shows that solid and liquid state NMR can be combined with mass spectrometry to provide information on such difficult systems.

As suggested, we have added perspective paragraph (lines 446-448).

Main article:

Abstract

suppress line 20-21, to fuse with line 18

... we propose a new model in which peptidoglycan cross-linking affects the activity, selectivity and specificity these two structurally related enzymes differently.

This has been corrected.

L62-63: (...), is the native species of Lss.

This has been corrected.

L87 Figure caption Fig 1: Please change this sentence: "schematic representation of S. aureus sacculus and LssCD (orange) and LytMCD (yellow) digestion site"

This has been corrected.

L92: authors claimed that the cross-bridges are indicated with an arrow, but this was not the case. Please update the caption or the figure accordingly.

This has been corrected.

Figure 2: During the second reviewing, I realize that the box area from the 2D NMR data in the ¹⁵N 118-128ppm and ¹H 7.6-8.2ppm differ significantly. Is the authors have any idea of the possible annotation of these peaks? Could be some evidences of cross digestion products by (e.g. other) M23 hydrolases in the spectra? Maybe these data contains additional evidences helping the interpretation of the structure-activity relationship.

This region has been annotated in previous work from the team (Maya-Martinez et al., 2019).

In our study, we are monitoring the glycine region, which is the one directly involved in enzymatic cleavage of the pentaglycine chain. It is also the most reliable region to assign by NMR, as it is obtained by the 3D heteronuclear experiments classically used for the assignment of the backbone chain. In the figure, we have now annotated the resonances corresponding to the lysine and alanine resonances located in the proximity of pentaglycine bridge. These resonances are, as expected, sensitive to hydrolysis reaction occurring inside the pentaglycine cross-bridge. Those peaks differ between peptidoglycan treated or not treated by M23 but no major differences are observed between the M23-treated samples. We have added this explanation in the legend caption (line 143-144).

What does the dark grey background mean in Fig S2c “muropeptides”?

This indicates “Glycines at acceptor stem-peptide” and is indicated on the figure. We have added this information in the figure caption.

Fig 2e: Even if we discard my previous comments in the beginning of my report: It would be better if the authors compared at least the absolute MS signals before and after the action of the LssCD (or LyMCD “enzymes), the heatmap would make more sense. Without any normalization, these data are meaningless.

In the present work, the only normalization relies on the sample preparation of the authors, which performed the bacterial growth until 0.6 O.D. Nonetheless, nothing ensures that the amount of produced full-length PG products or degradation products are produced at the same level between experiments, unless the authors produced only one big batch of PG extract. In this case, it should be claimed by the authors in the M&M.

Indeed, we used the same batch and amount of PG extract for all presented digestion experiments. We have included this information in M&M (line 504, 506-508).

Following the strategy used by the reviewer to present MS data, we changed the figures 2e and 3c accordingly.

L172-177: Authors claimed that LytMCD does not release soluble products. Nonetheless LytMCD releases oligoglycines that were not monitored by NMR or MS, which are soluble released products. *In the referred lines we write: “no reaction products of sacculus hydrolysis **could be detected** after treatment with LytMCD”, not that these are not released, which we find in agreement with reviewers’ reasoning.*

L180: Proposition of rephrasing. “We then hypothesized that LytMCD could cleave the cross-bridge without releasing soluble fragments.”

This has been corrected.

L200: Please remind to the reader that some of your mass spectra are deconvoluted spectra while other are raw (zoomed) mass spectra. I am a MS expert, so it is obvious for my eyes but it is not for most of your target readers.

This has been added to the caption of supplementary figure 2a.

L202: Due to the use of an excess of NaBH₄ for the reduction of your PG fragment (because of the CMS analysis later), the amount of Na (and K) adducts will be still very significant, even if the samples were injected in an LC column before MS detection. Consequently this will strongly affect the result of (semi) quantitation of the different targets by MS.

We revised this section and indeed observed Na⁺ and K⁺ adducts in analysis of substrates and products. Their intensities were extracted, added and sum of all intensities was presented in table S2, and included in final calculations presented on figure 2e and 3c.

L204-206: I think instead that if the cross linking of the PG decrease, then the amount of especially monomers, as well as dimers, and trimers would increase.

To the contrary if the cross-linking increase, then the amount of monomers, dimers, and trimers will decrease. I suggest to revisit this section.

*In this analysis we focused on the analysis of primary cross-linking products, which results in the formation of up to pentameric components catalysed by PBP2 (and PBP2A) (reference: PMID: 15716453, PMID: 28691491). PBP2 is the target of oxacillin (manuscript reference no. 43). Therefore, we concluded that decrease of dimers and trimers would validate decreased cross-linking in *S. aureus* treated with sub-inhibitory concentration of oxacillin. Nevertheless, following previous suggestions of reviewer concerning normalization of data (lack of internal standard) and necessity to use two different PG pellets in this analysis (wild-type and sub-MIC oxacillin treated culture), we decided to remove this analysis from the supplementary material (fig. S3). Since the*

effect of sub-inhibitory concentration of beta-lactam antibiotics on PG crosslinking in S. aureus is firmly documented in literature (manuscript references 44-46), we decided to provide literature background as a base for our further conclusions. We included necessary literature references in the text (line 184-187).

L256: I am still requesting to the authors to explain/justify the use of S. aureus SH1000 strain. What is the specificity/properties of the PG produced by this strain? Please provide some context somewhere in the introduction, or M&M or the plain text.

The context for using this strain has been provided in M&M (line 537-539).

L305/ Please mention in the figure caption that the residues are numbered following the nomenclature provided in Figure S7.

This has been added.

L412: Please correct that the NMR spectra are ^1H ^{13}C ^{15}N labelled LssCD instead of ^2H ^{13}C ^{15}N labeled LssCD.

For ssNMR proteins has been deuterated to improve quality of spectra, which allows exchangeable amide proton detection in ^1H , ^{15}N -correlation experiments. The deuteration procedure is provided in M&M (line 472 and 595).

L438: Please mention that you (probably) refer to the Fig. 3C (multimeric forms of muuropeptides are digested efficiently by LssCD but not by LytMCD).

This has been added.

L548: I assumed that you used a Synapt G2 HDMS. If not please specify which series of SynaptG2 you used.

This has been added to M&M.

L553: 650 L/h, not l/h

This has been corrected.

Supporting information :

Disclaimer:

During the reviewing process, the reviewer does not have any access to MS software tools. All was manually processed and mistakes cannot be excluded.

Figure S1 : For clarity, I would better suggest to edit the figure caption as « Chemical structure of the non cross-linked peptidoglycans and oligoglycin peptides (...) »

Note that neither chemical formula or masses or provided, in contrast with what is claimed in this caption.

If the authors want to present MS data, chemical formula is a more useful information than « masses ». Indeed we do not know what kind of masses are provided : neutral compounds, ions, adducts, and charge state ... ?

Caption has been rephrased accordingly. For MS data (fig. S2) chemical formula has been presented instead of masses.

Figure S2 : The exact mass I computed differ from the mass provided by the authors. Moreover the m/z provided by the authors change for the same species. Please provide uniform information.

Consequently one can deduce that the average mass accuracy of the MS data is about 50ppm (RMS), which is surprisingly high for a Synapt G2 mass spectrometer, even for external mass calibration.

This is suggesting that the intensities of the NaI calibrant do not fit the intensity of the ions detected during UPLC MS experiments.

If I reinterpret the MS mass spectrum provided in SI, I can deduce species having 1 Da mass lower than the mass suggested by the authors (mono-isotopic mass would be 626,39 instead of 627,7x presented by the authors, $z=+2$).

Additionally, I have to assume that the MS instrument is actually a Synapt G2 HDMS instrument. So please, explain how the authors determined the mass $[M+H]_{x+}$ presented in the SI.

In my opinion, authors should provide the theoretical exact mass of the monitored species in regards of the showed mass spectra.

Figure S3: Globally same remarks

We have analysed the raw data that we received from MS service (Mass Spectrometry Laboratory, IBB PAS). The mass spectrometer used there is Synapt G2 MS (Waters). Mass was calculated by MaxEnt3 in MassLynx 4.1 software. The mean m/z value presented above m/z spectra were calculated manually based on m/z value and peak intensity, which is not an exact calculation. The instrument calibration and the lack of internal Lock-Spray (e.g. Leu-Enkephalin commonly used during HDX-MS analysis) led also to discrepancies in m/z values. Moreover, sometimes we observe ion interference and overlapping peaks leading to poor mass measurement accuracy.

Following reviewers' suggestions, we are providing now uniform information for m/z spectra, chemical formulas and theoretical exact masses of the monitored species.

Figure S4: Provide the signification of MHB in the SI

This has been added.

Figure S14 and S15: The residues and their respective numerotation are barely readable. This is making hard for the reader to conceive how would/could be the interaction between the residues of the enzyme and the substrate, especially in mentally reconstructed 3D structures.

The quality of these figures has been improved.

Table S1 and S2 have a typo in « muropeptides »

This has been corrected.

REVIEWERS' COMMENTS

Reviewer #1 (Remarks to the Author):

Razew and coworkers, *S. aureus* sacculus mediates activities of M23 hydrolases

NCOMMS-23-16038B, Nature comm., revision 3

Once again, I appreciated the efforts of the authors for complying with my revision requests.

I believe that this last version should make easier to read and understand for the readers which are not NMR experts or MS experts.

Concerning the detection of the soluble products:

You answered me about the “(...) no reaction of sacculus hydrolysis could be detected after treatment with LytMCD”.

Here my concern was semantic. We cannot ever detect something that the detector is not capable to detect, for example because the mass range of MS detector does not fit with the m/z of such targets. Consequently the non detection does not mean that the targets were not present in the samples.

I think the most appropriate way to rephrase the sentence could be something like “Under our experimental conditions, no reaction products (...) could be detected after treatment (...)”

I also appreciated the mitigation of the conclusion and that the authors suggest to validate the model with orthogonal molecular or biophysical methods.

The supplementary information were also revised accordingly.

I am still not sure about the choice of the *S. aureus* SH1000 strain. I think, but still not sure, that SH1000 was selected because you (and the community) have some accumulated knowledge and background of this strain, and not because of e.g. a previously and exhaustively characterized cross-linking of the peptidoglycane, or almost “monodisperse” or homogeneous reticulation of the peptidoglycans, or whatever else.

Consequently, at this stage of the reviewing process, I have no more complain and I recommend the publication of the article after some very minor modifications.

REVIEWERS' COMMENTS

Reviewer #1 (Remarks to the Author):

Razew and coworkers, *S. aureus* sacculus mediates activities of M23 hydrolases
NCOMMS-23-16038B, Nature comm., revision 3

Once again, I appreciated the efforts of the authors for complying with my revision requests. I believe that this last version should make easier to read and understand for the readers which are not NMR experts or MS experts.

Concerning the detection of the soluble products:

You answered me about the “(...) no reaction of sacculus hydrolysis could be detected after treatment with LytMCD”.

Here my concern was semantic. We cannot ever detect something that the detector is not capable to detect, for example because the mass range of MS detector does not fit with the m/z of such targets. Consequently the non detection does not mean that the targets were not present in the samples.

I think the most appropriate way to rephrase the sentence could be something like “Under our experimental conditions, no reaction products (...) could be detected after treatment (...)”

This has been included in the text.

I also appreciated the mitigation of the conclusion and that the authors suggest to validate the model with orthogonal molecular or biophysical methods.

The supplementary information were also revised accordingly.

I am still not sure about the choice of the *S. aureus* SH1000 strain. I think, but still not sure, that SH1000 was selected because you (and the community) have some accumulated knowledge and background of this strain, and not because of e.g. a previously and exhaustively characterized cross-linking of the peptidoglycane, or almost “monodisperse” or homogeneous reticulation of the peptidoglycans, or whatever else.

This is exactly the case. We emphasised this point by providing references to SH1000 strain in Methods section to validate it is a model, laboratory strain.

Consequently, at this stage of the reviewing process, I have no more complain and I recommend the publication of the article after some very minor modifications.

We thank Reviewer for positive feedback on our work and appreciate his/her effort for ameliorating the submitted manuscript.